# *Caenorhabditis elegans* sperm carry a histone-based epigenetic memory of both spermatogenesis and oogenesis

Tomoko M. Tabuchi[1], Andreas Rechtsteiner[1], Tess E. Jeffers[2], Thea A. Egelhofer[1], Coleen T. Murphy[2] & Susan Strome [1]

Paternal contributions to epigenetic inheritance are not well understood. Paternal contributions via marked nucleosomes are particularly understudied, in part because sperm in some organisms replace the majority of nucleosome packaging with protamine packaging. Here we report that in *Caenorhabditis elegans* sperm, the genome is packaged in nucleosomes and carries a histone-based epigenetic memory of genes expressed during spermatogenesis, which unexpectedly include genes well known for their expression during oogenesis. In sperm, genes with spermatogenesis-restricted expression are uniquely marked with both active and repressive marks, which may reflect a sperm-specific chromatin signature. We further demonstrate that epigenetic information provided by sperm is important and in fact sufficient to guide proper germ cell development in offspring. This study establishes one mode of paternal epigenetic inheritance and offers a potential mechanism for how the life experiences of fathers may impact the development and health of their descendants.

[1] Department of Molecular, Cell and Developmental Biology, University of California Santa Cruz, 1156 High Street, Santa Cruz, CA 95064, USA. [2] Department of Molecular Biology and LSI Genomics, Carl Icahn Lab 148, Princeton University, Princeton, NJ 08545, USA. Correspondence and requests for materials should be addressed to S.S. (email: sstrome@ucsc.edu)

Epidemiological studies in humans and experiments in mammalian models have revealed that conditions experienced by fathers can affect future generations[1,2]. However, the mechanisms by which fathers transmit information beyond the DNA code, in other words epigenetic information, to future generations are not well understood. Paternal contributions via chromatin marking are especially mysterious. This is partly because sperm DNA in some organisms is repackaged with protamines and with reduced levels of histones, complicating analysis of the sperm epigenome and challenging the notion that marked histones in sperm may provide epigenetic memory. Advances in genomics have enabled researchers to decipher the epigenetic landscape of sperm from humans, mice, and zebrafish[3–7]. Those studies demonstrated that the sperm genome retains histones, although the extent varies from 1–10% in mammals to 100% in zebrafish. Retained histones are modified with active and/or repressive histone modifications, unveiling the potential for sperm to transmit epigenetic information to offspring.

The presence of modified histones in sperm raises important questions. Can sperm transmit epigenetic information to offspring in the form of histone modifications? What is the fate of sperm-inherited marking in early embryos? Does sperm-inherited marking impact offspring development? *C. elegans* offers an exceptional system in which to address these questions. *C. elegans* lacks DNA methylation on cytosines[8], one established mediator of epigenetic control, and thus may rely more heavily on histone modifications to transmit epigenetic information. Indeed, *C. elegans* sperm retain at least some histone packaging of the genome[9–11], deliver chromosomes marked with histone modifications to embryos[9–11], and can transmit a chromatin-based memory of temperature, diet, and stress to offspring[12–14]. Here we analyzed the epigenome of *C. elegans* sperm and tested whether epigenetic information in the form of marked nucleosomes influences germline development in offspring.

Here we document that *C. elegans* sperm retain modified histones genomewide, similar to zebrafish sperm, and that sperm carry a histone-based epigenetic memory of genes with spermatogenesis-restricted expression and unexpectedly also genes with oogenesis-enriched expression. In sperm, spermatogenesis-restricted genes are marked with an unusual combination of active and repressive histone modifications, which may be a sperm-specific signature and which resolves to retention of only repressive marks in early embryos. We find that genes previously shown to have enriched expression during oogenesis are also transcribed during spermatogenesis and thus bear active marks as a result of their transcription. We demonstrate that sperm marking is important for normal germ cell development in offspring whose germ cells inherit both sperm and oocyte chromosomes, and sufficient for normal germ cell development in offspring whose germ cells inherit only sperm chromosomes. Our studies establish modified histones retained in mature sperm as one mechanism by which fathers may transmit heritable traits, and highlight a role for sperm epigenetics in the development and fertility of descendants.

## Results

*C. elegans* sperm retain nucleosomes genome-wide. To determine how the genome is packaged in *C. elegans* sperm, we isolated adult males and collected mature sperm (~ 99% purity, Supplementary Fig. 1a). We utilized micrococcal nuclease digestion followed by paired-end sequencing (MNase-seq) to evaluate the presence of nucleosomes across the genome in sperm. We found that sperm retain nucleosomes across all six chromosomes, comparable in presence and gross distribution to early embryos but with half the MNase-seq reads from the X chromosome in sperm compared to early embryos, as expected based on their different X: A ratios (Supplementary Fig. 2a-b). Genome-wide retention of nucleosomes in *C. elegans* sperm resembles zebrafish sperm and differs from mammalian sperm, which replace 90–99% of their histone packaging with protamine packaging[3–7].

To analyze the distribution of histone modifications across the genome in *C. elegans* sperm, we solubilized formaldehyde-fixed mononucleosomes from sperm and performed chromatin immunoprecipitation followed by sequencing (ChIP-seq). We focused on a modification associated with gene repression (trimethylation of histone H3 on Lys27, H3K27me3) and two modifications associated with gene expression (H3K36me3 and H3K4me3), because these modifications have been implicated in transgenerational epigenetic inheritance in *C. elegans*[9,11,14–18]. We found that *C. elegans* sperm retain modified histones across all six chromosomes with alternating H3K36me3- and H3K27me3-marked chromatin domains across the five autosomes (Fig. 1, Supplementary Fig. 2c). This domain organization is similar to that observed in early embryos[15,19]: H3K36me3 domains span the coding regions of single genes or sets of adjacent genes, and H3K27me3 domains overlie silent genes and intergenic regions. The level of H3K4me3 was lower in sperm compared with early embryos. We conclude that in *C. elegans* sperm the genome is packaged with nucleosomes bearing histone modifications.

To gain a comprehensive view of the epigenome transmitted to embryos by both gametes, we profiled oocyte chromatin using ChIP-seq analysis. As wild-type oocytes cannot be obtained in sufficient quantities for ChIP-seq, we profiled chromatin from ovulated but unfertilized oocytes collected from feminized worms. These oocytes were estimated to be 90–95% pure with slight contamination from gonads and immature oocytes (Supplementary Fig. 1b, Methods). Ovulated but unfertilized oocytes from feminized worms were previously shown to display changes in RNA accumulation compared to wild-type oocytes[20]. We observed similar changes with our oocyte preparations from feminized worms and determined those changes to be mainly downregulation of germline genes and upregulation of somatic genes, typical of an embryogenesis program. Nevertheless, the chromatin in those oocytes still displays histone marking consistent with oocytes: 84% of genes expressed during oogenesis (called "oogenesis genes", ref. [21] and Methods) bear the active mark H3K36me3, and 86% bear the active mark H3K4me3 (Supplementary Fig. 3d, e); 81% of soma-specific genes (Methods) and 83% of strict embryo genes[22] bear the repressive mark H3K27me3 (Supplementary Fig. 3d, e). We conclude that ChIP-seq data from our oocyte preparations largely reflect oocyte chromatin states.

**Sperm display multivalent marking on spermatogenesis genes.** Comparison of ChIP-seq data revealed that sperm, oocytes, and early embryos display very similar chromatin domains marked with either H3K36me3 or H3K27me3 (Fig. 1b, Fig. 2a–c, the second ChIP-seq replicate is shown in Supplementary Fig. 4). To ask whether genes are marked with H3K36me3, H3K27me3, both, or neither, we calculated the mean normalized ChIP signal for each protein-coding gene over the gene body and displayed the values in scatter plots. In all three cell types, sperm, oocytes, and early embryos, the majority of genes appear in the top-left and the bottom-right quadrants, as expected, showing that 86% of all genes are marked with either H3K27me3 or H3K36me3 (Supplementary Fig. 5, gray). To correlate marking of genes with their transcription status in adult germlines, we profiled by RNA sequencing (RNA-seq) the transcriptome of spermatogenic and

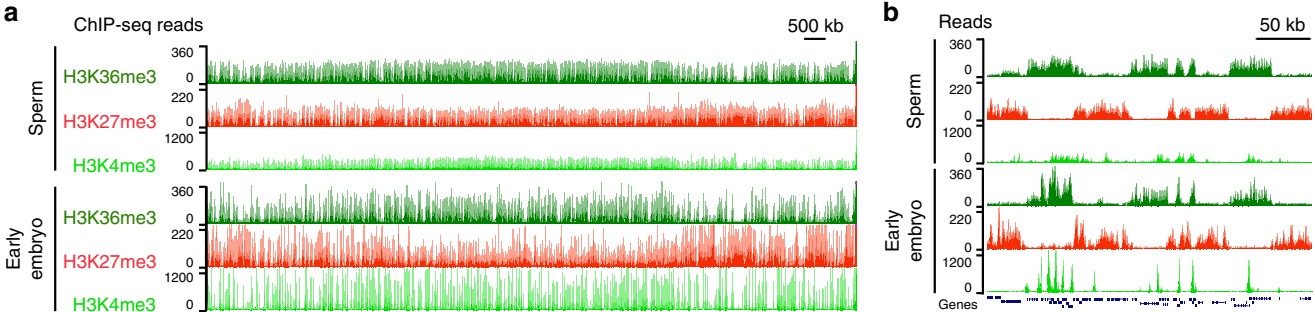

**Fig. 1** *C. elegans* sperm retain modified histones across the genome, and H3K36me3 and H3K27me3 generally occupy mutually exclusive domains. **a** Genome-browser view of H3K36me3 (green), H3K27me3 (red), and H3K4me3 (lime) ChIP-seq from sperm and early embryos across chromosome l. The y-axis shows normalized ChIP-seq read counts. **b** Zoomed-in genome-browser view, showing the generally mutually exclusive occupancy of H3K36me3 and H3K27me3 in sperm and early embryos over ∼ 40 genes. The entire genome is shown in Supplementary Fig. 2c. Figs 1–3 and Supplementary Fig. 5 show 1 replicate. Another biological replicate, shown in Supplementary Fig. 4, demonstrates the consistency of the data

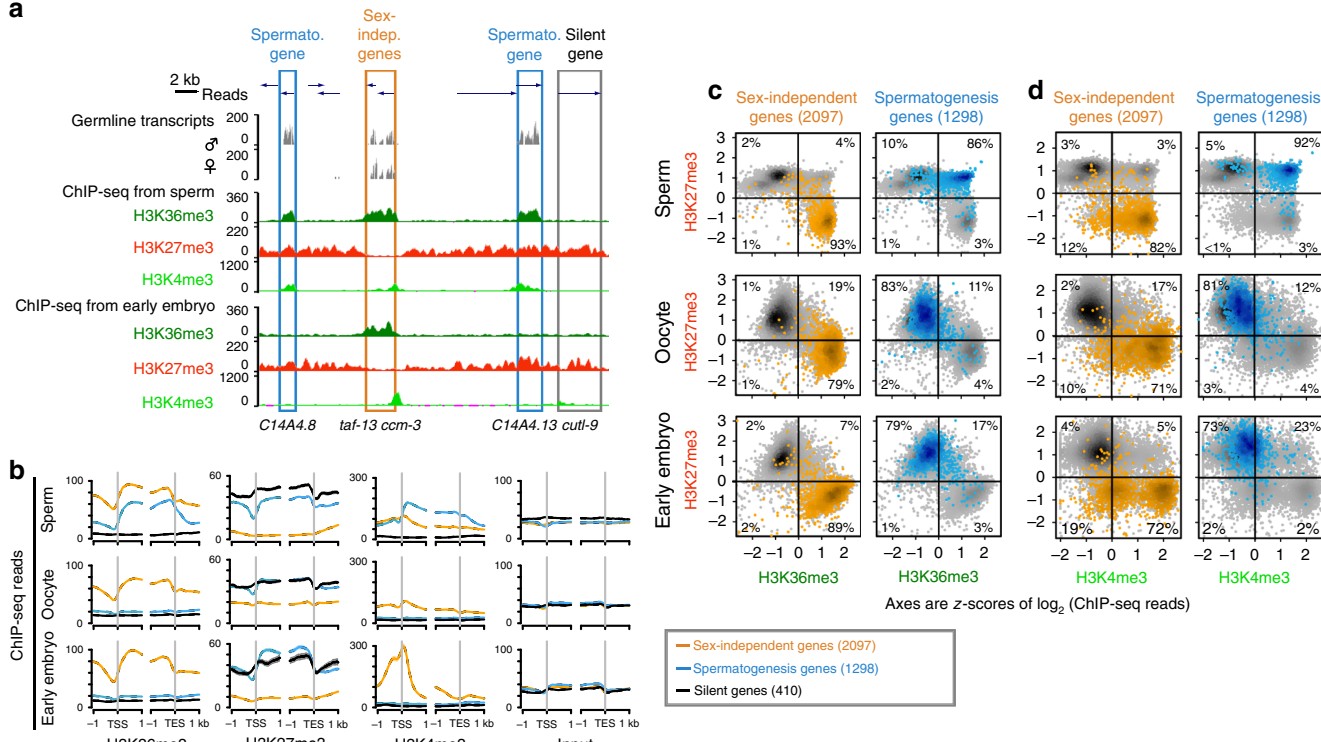

**Fig. 2** Spermatogenesis genes are marked with both active and repressive histone modifications in sperm. **a** Genome-browser views of RNA-seq data from spermatogenic and oogenic germlines, and ChIP-seq data (H3K36me3, H3K27me3, and H3K4me3) from sperm and early embryos. Blue: genes transcribed exclusively in spermatogenic germlines ("spermatogenesis genes", Methods). Gold: genes transcribed in both spermatogenic and oogenic germlines ("sex-independent genes", Methods). Open-box: gene not transcribed in spermatogenic or oogenic germlines. The y-axis shows normalized ChIP-seq read counts. **b** Normalized ChIP signals around (1 kb up- and downstream) the transcript start site (TSS) and transcript end site (TES) over sex-independent, spermatogenesis, and silent genes in sperm, oocytes, and early embryos. **c**, **d** Scatter plots of mean normalized gene-body H3K36me3 (**c**) or promoter H3K4me3 (**d**) vs. gene-body H3K27me3 ChIP signals for all coding genes (gray), highlighting sex-independent genes (gold) and spermatogenesis genes (blue) and noting the % of genes in each quadrant. Classes of genes are shown in separate panels in Supplementary Fig. 5

oogenic germlines. We found that genes expressed in both spermatogenic and oogenic germlines (called "sex-independent genes", Methods) are generally marked with the active modification H3K36me3 and lack the repressive modification H3K27me3 (Fig. 2a–c). Genes that are not transcribed in either spermatogenic or oogenic germlines (called "silent genes", ref. [23] and Methods) are marked with the repressive modification H3K27me3 only (Fig. 2a, b). In short, genes expressed in both spermatogenic and oogenic germlines are marked with H3K36me3, and genes silent in both are marked with H3K27me3,

following an "either H3K36me3 or H3K27me3" rule, consistent with the known antagonism between H3K36me3 and H3K27me3[24,25].

A surprising exception to the "either H3K36me3 or H3K27me3" rule is spermatogenesis genes in sperm. These are genes expressed exclusively in spermatogenic germlines and not in oogenic germlines ("spermatogenesis-specific genes" or "spermatogenesis genes" for short, Supplementary Fig. 3a, Methods). Strikingly, these genes (as well as previously defined sperm-enriched genes[21,26]) display *both* H3K36me3 and

H3K27me3 over their gene body in sperm (Fig. 2a–c). Also surprising is the presence of another active histone modification, H3K4me3, over the body of spermatogenesis genes in sperm (Fig. 2a, b, d); that mark typically appears as a sharp peak at the promoter of expressed genes. Taken together, our data suggest that in sperm spermatogenesis genes are marked with both active (H3K36me3 and H3K4me3) and repressive (H3K27me3) marks, which we refer to as "multivalent" marking. In oocytes, spermatogenesis genes bear only H3K27me3 as expected, as they are not expressed during oogenesis (Fig. 2b–d). In early embryos, spermatogenesis genes also bear only H3K27me3 (Fig. 2), suggesting that the two active modifications transmitted from sperm are lost or erased in early embryos. We conclude that sperm carry a history of spermatogenesis gene expression using an unusual combination of histone modifications.

**Sperm display active histone marks on oogenesis genes.** Another class of genes with unexpected marking in sperm is genes with oogenesis-enriched expression ("oogenesis-enriched genes" or "oogenesis genes" for short[21,26]; Methods). Some gene products from this class are directly involved in oogenesis (e.g., the yolk receptor RME-2), whereas others are maternally synthesized and oocyte-supplied gene products involved in early embryo development (e.g., EGG-4, PIE-1, PAR-6, and NOS-2). We expected these genes to be transcriptionally repressed in male germlines and to bear H3K27me3 in sperm. Unexpectedly, we found that in sperm 78% of oogenesis genes are instead marked with H3K36me3 and 75% with H3K4me3, and are devoid of H3K27me3, typical of expressed genes (Fig. 3a–c, Supplementary Fig. 3c–e). We consider two possible mechanisms by which such oogenesis genes exhibit active epigenetic states in sperm. Marking of oogenesis genes is: (1) maternally inherited and maintained in males or (2) generated de novo in male germlines in a transcription-dependent manner. To test the latter, we analyzed our germline RNA-seq data and found that 73% of oogenesis genes (82% of those bearing the active mark H3K36me3 in sperm) are indeed transcribed in spermatogenic germlines (i.e., have Reads Per Kilobase of transcript per Million mapped reads (RPKM) values > 15 in Fig. 3d, Methods). As an independent method to test this and to visualize spatiotemporal patterns of expression, we employed single-molecule fluorescence in situ hybridization (smFISH)[27], using probes against well-known oogenesis messenger RNAs (pie-1, par-6, nos-2, mex-3, and cpg-2). smFISH indeed detected transcripts from these genes in spermatogenic germlines, especially during pachytene (Fig. 3e, Supplementary Fig. 6). As expected, oogenesis genes display active histone modifications in oocyte ChIP-seq and show smFISH signal in oogenic germlines (Fig. 3, Supplementary Fig. 3d,e, Supplementary Fig. 6). Our observation that many oogenesis genes are transcribed in male germlines is consistent with other published findings: (1) during spermatogenesis in hermaphrodites, the RNA-binding proteins FOG-1 and FOG-3, which promote sperm cell fate, predominantly associate with oogenic transcripts[28], and (2) males contain many small RNAs that guide the activity of the Argonaute protein CSR-1 and that are antisense to oogenic transcripts[29]. Indeed, we were not able to identify a high-confidence set of "oogenesis-specific genes" (i.e., expressed in oogenic germlines and not in spermatogenic germlines) using the same criteria we used to identify "spermatogenesis-specific genes" (Supplementary Fig. 3c, Methods). Thus, most oogenesis genes are transcribed in male germlines, perhaps to maintain those genes in an open chromatin state in sperm. We conclude that sperm carry a histone-based epigenetic memory of spermatogenesis gene expression, which includes (1) spermatogenesis genes bearing multivalent marking and (2) genes well known for their expression during oogenesis bearing active marking.

**Sperm chromatin is important for the fertility of offspring.** Our observation that sperm chromosomes are marked in a manner that reflects germline gene expression prompted the question whether sperm chromatin marking is important for germline development in offspring. To address this, we analyzed the fertility of worms that inherited sperm chromosomes lacking a histone modification thought to be important for epigenetic memory. The three histone modifications that have been implicated in epigenetic memory in *C. elegans* are methylated H3K4, H3K36, and H3K27[9,11,14–18]. The first two marks each rely on multiple histone methyltransferases[15,16] and are challenging to completely deplete from sperm. Consequently, we focused on H3K27me3, which can be eliminated from sperm chromosomes by disrupting Polycomb Repressive Complex 2 (PRC2) in males[11,30]. Males homozygous for a null mutation in *mes-3*, which encodes a subunit of the worm PRC2 complex, were mated with feminized *mes-3/+* heterozygous worms to produce offspring that inherited sperm chromosomes lacking H3K27me3 (Fig. 4a). High percentages of the resulting *mes-3* homozygous (92%) and heterozygous (73%) offspring developed into sterile adults in this sensitized genetic background (Fig. 4b, red, M + P- for Maternally marked chromosomes and Paternal chromosomes lacking marking, Methods). In contrast, genetically identical control offspring that received H3K27me3-marked sperm chromosomes displayed low sterility (Fig. 4b, blue, M + P + ). This suggests that paternally transmitted epigenetic information is important for germline development in offspring.

We next analyzed gene expression changes in the germlines of M + P − offspring that show high sterility and investigated whether they reflect sperm transmission of altered chromatin states or an altered cargo of mRNAs. The germlines of M + P − offspring displayed mainly upregulation of somatic genes and genes on the X chromosome, as previously documented in M + Z − (Maternal load of MES protein, no Zygotic synthesis) *mes* mutant germlines[24]. We compared the transcriptomes of *mes-3* male germlines (parent germlines), *mes-3* sperm, and germlines from M + P − offspring, along with their respective controls (Fig. 4c, d). We found significant overlaps between genes with altered mRNA accumulation in parent germlines and in sperm (p < 10^−70, hypergeometric test in R), and between genes upregulated in parent germlines and in germlines in M + P − offspring (p < 10^−5) (Fig. 4c). We did not observe a significant overlap between genes with altered mRNA accumulation in sperm and in germlines in M + P − offspring (p = 0.4) (Fig. 4c). These findings suggest that at least some genes upregulated in parent germlines are also upregulated in the germlines of M + P − offspring, and that an altered cargo of mRNAs in mutant sperm does not mediate this. This is further supported by density plot analyses of three sets of genes identified in these RNA-seq experiments: 123 genes upregulated in *mes-3* parent germlines and in *mes-3* sperm, 439 genes upregulated in *mes-3* parent germlines but not in *mes-3* sperm, and 81 genes upregulated in *mes-3* sperm but not in *mes-3* parent germlines (Fig. 4d). The two gene sets upregulated in parent germlines (plus or minus sperm) were statistically significantly upregulated in the germlines of M + P − offspring when compared with all genes, whereas the gene set upregulated only in sperm was not significantly upregulated in the germlines of M + P − offspring. These findings suggest that *mes-3* mutant sperm convey a memory of gene upregulation to the next generation not via an altered payload of mRNAs but perhaps via altered chromatin states.

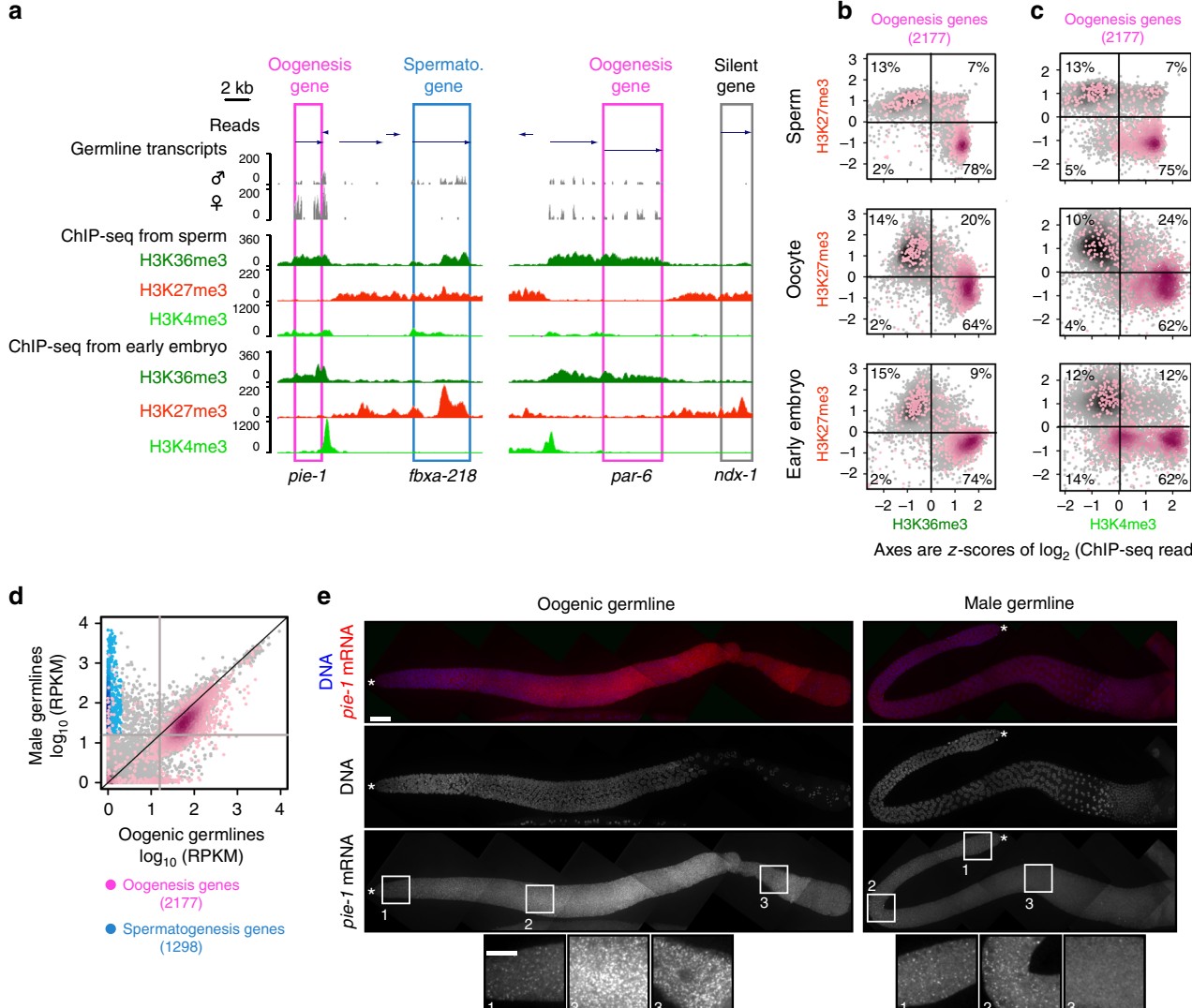

**Fig. 3** "Oogenesis genes" are marked with active modifications in sperm and are transcribed in male germlines. **a–c** Genome-browser views and scatter plots as described for Fig. 2, highlighting genes defined as "oogenesis genes" (Methods) in pink. Classes of genes are shown in separate panels in Supplementary Fig. 5. **d** Comparison of spermatogenic vs. oogenic germline transcriptomes, highlighting spermatogenesis genes (blue) and oogenesis genes (pink). Axes show log-transformed RPKMs after adding a pseudo-count of 1. The gray lines show the cutoff for expressed genes (RPKM of 15, Methods). The number of biological replicates for spermatogenesis germlines was 6 and for oogenesis germlines was 3. **e** smFISH images of RNA from a well-known oogenesis gene *pie-1* in oogenic and male germlines. Top panels are z-stack projections with the distal tip marked with *. Bottom panels are high-magnification views of the boxed regions (projection of 3 slices). For both oogenic and male germlines, 10–20 gonads were visually examined under the microscope and at least 3 germlines were imaged. Scale bars represent 20 µm for the top panels and 10 µm for the bottom panels

**Sperm chromatin is sufficient for the fertility of offspring.** Given that sperm chromosomes are marked in a manner that reflects both spermatogenic and oogenic gene expression patterns, that epigenetic marking of sperm chromosomes is faithfully transmitted through embryo cell divisions[11], and that sperm epigenetic marking is important in offspring, we next tested whether sperm epigenetic marking alone is sufficient for proper development of the germline in offspring. We utilized a mutant that, during the first embryonic division, delivers the sperm genome to the daughter cell that generates the germline and the oocyte genome to the other daughter cell[31]. This mutant over-expresses GPR-1, a protein involved in regulation of kinetochore pulling forces. GPR-1 overexpression results in excessive pulling forces, causing the paternal and maternal pronuclei to inappropriately move to opposite poles of the one-cell embryo instead of merging in the center of the embryo. In this mutant background, ~ 60% of offspring undergo atypical chromosome segregation,

generating mosaic embryos whose germlines are derived entirely from sperm chromosomes, and ~ 40% of offspring undergo normal chromosome segregation[31]. To track the parental genomes, differentially tagged histone transgenes were used[31]: a green fluorescent protein (GFP)-tagged histone H2B encoded in the sperm genome and a TdTomato-tagged histone H2B encoded in the oocyte genome (Fig. 5a). Remarkably, we found that mosaic embryos whose germline inherited only sperm chromosomes ("red-head" worms) develop into fertile adults (as observed in ref. [31]) with a normal brood size, similar to control worms ("yellow" worms) in which the germline inherited both sperm and oocyte chromosomes (Fig. 5b). RNA-seq analysis demonstrated that the germline transcriptome of "red-head" worms and their offspring show few (< 80 genes) and minor changes compared with control worms (Fig. 5c). These findings demonstrate that epigenetic information provided by sperm can guide proper germ cell development. Our findings are consistent with those of

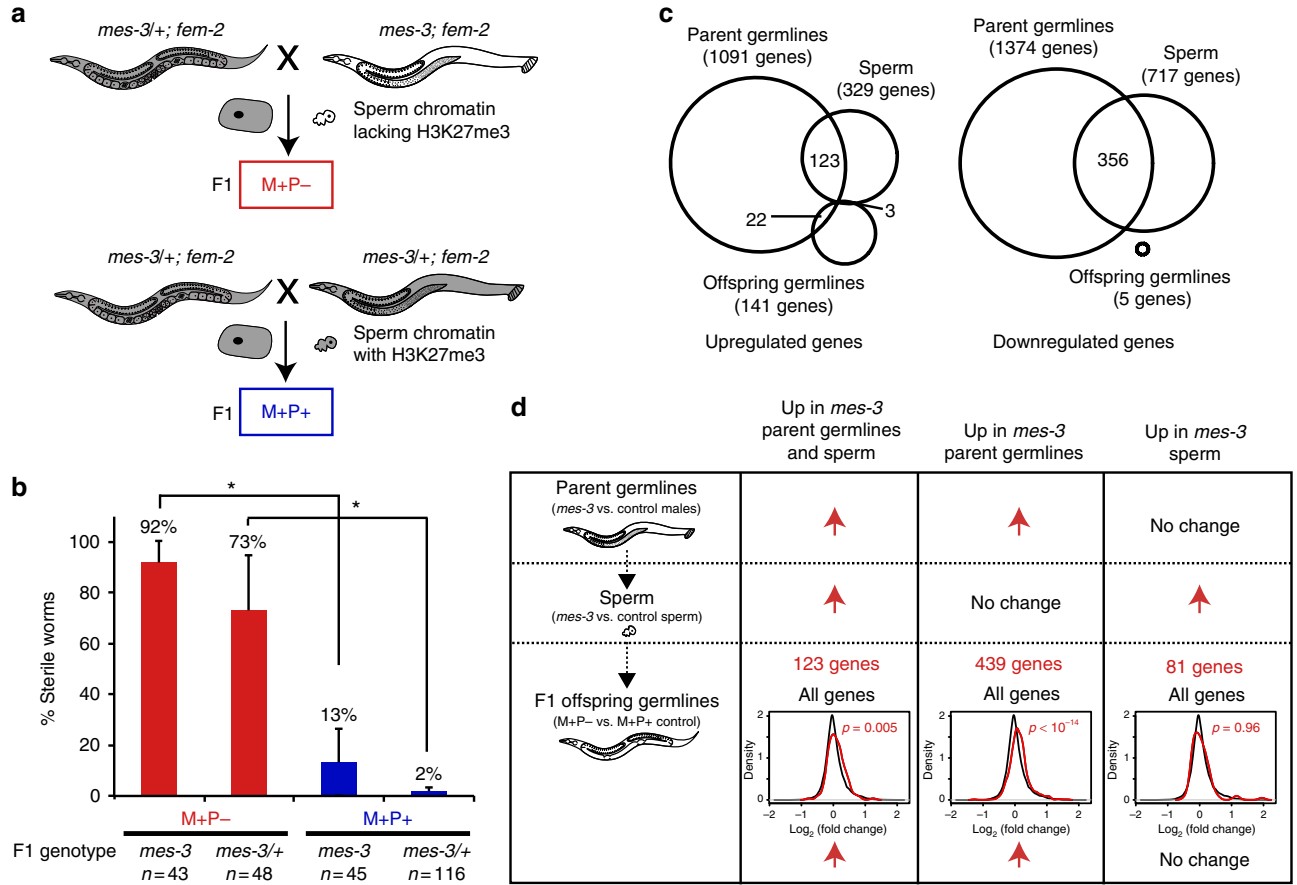

**Fig. 4** The sperm epigenome is important for germline development in offspring. **a** Genetic strategy to test if sperm chromatin marking is necessary for germ cell development in offspring that inherit both sperm and oocyte chromosomes, using male parents that either can or cannot generate H3K27me3. **b** Histogram showing % sterile hermaphrodites among the F1 offspring that received sperm chromosomes lacking H3K27me3 (red, M + P − for Maternally marked chromosomes, Paternal chromosomes lacking the mark) vs. properly marked sperm chromosomes (blue, M + P + ). Error bars show SD of 3 experiments. * Statistically significant differences between the genetically identical M + P − vs. M + P + offspring, one-tailed Student's *t*-test, $p \leq 0.01$. **c** RNA-seq analysis of germlines from male parents, mature sperm, and the germlines of F1 offspring. Venn diagrams show genes significantly up- or downregulated in *mes-3* male germlines (3 biological replicates) compared with control male germlines (4 biological replicates), *mes-3* sperm compared with control sperm (3 biological replicates each), and F1 M + P − germlines compared with genetically identical F1 M + P + germlines (2 biological replicates each). A hypergeometric test in R demonstrated significant overlaps between genes misregulated in parent germlines and in sperm ($p < 10^{-70}$) and between genes upregulated in parent germlines and in germlines in M + P − offspring ($p < 10^{-5}$). No significant overlap was observed between genes misregulated in sperm and in germlines in M + P − offspring ($p = 0.4$). **d** Density plots showing the distribution of $\log_2$ (fold change) of transcripts from all genes (black) and subsets of genes (red) in F1 M + P − germlines compared to control F1 M + P + germlines. 123 genes were significantly upregulated in *mes-3* male germlines (red arrow) and in their sperm (red arrow). These genes as a set (red, left) were statistically significantly upregulated in F1 M + P − germlines when compared with all genes (black) ($p = 0.005$). 439 genes were significantly upregulated in *mes-3* male germlines but not in their sperm. This gene set (red, middle) was also statistically significantly upregulated in F1 M + P − germlines when compared with all genes (black) ($p < 10^{-14}$). 81 genes were significantly upregulated in *mes-3* sperm but not in parental male germlines. This gene set (red, right) was not significantly upregulated in F1 M + P − germlines when compared with all genes (black) ($p = 0.96$). Unpaired Mann–Whitney *U*-test in R was used to calculate *p*-values for density plots. DESeq2 was used to calculate false discovery rate (FDR) of differential gene expression: FDR < 0.05 significantly different, FDR ≥ 0.2 not significantly different

ref. [32] that showed that offspring that inherit both homologs of a marked autosome from the oocyte or the sperm are viable and fertile, indicating that *C. elegans* autosomes lack imprinting. Indeed, it appears likely that either oocyte-derived or sperm-derived chromosomes can support normal development[31,32].

## Discussion

*C. elegans* sperm resemble zebrafish sperm in retaining nucleosomes genome-wide. In contrast, mammalian sperm replace 90–99% of their nucleosome packaging with protamine packaging. We note that protamine-like proteins have been identified in *C. elegans*[10,33]. It is possible that in *C. elegans* sperm some regions are packaged with a mixture of histones and protamines, as suggested by analysis of mammalian sperm[3,7,34].

Multivalent marking of spermatogenesis-specific genes in *C. elegans* sperm reflects colocalization by ChIP of the assayed histone modifications, the active marks H3K4me3 and H3K36me3 and the repressive mark H3K27me3. There are at least three possible explanations for this colocalization by ChIP: (1) active and repressive marks coexist on the same H3 tails, (2) active and repressive marks exist on different H3 tails of the same nucleosomes, and (3) active and repressive marks exist on different nucleosomes in the population analyzed. Further studies are required to distinguish between these possibilities.

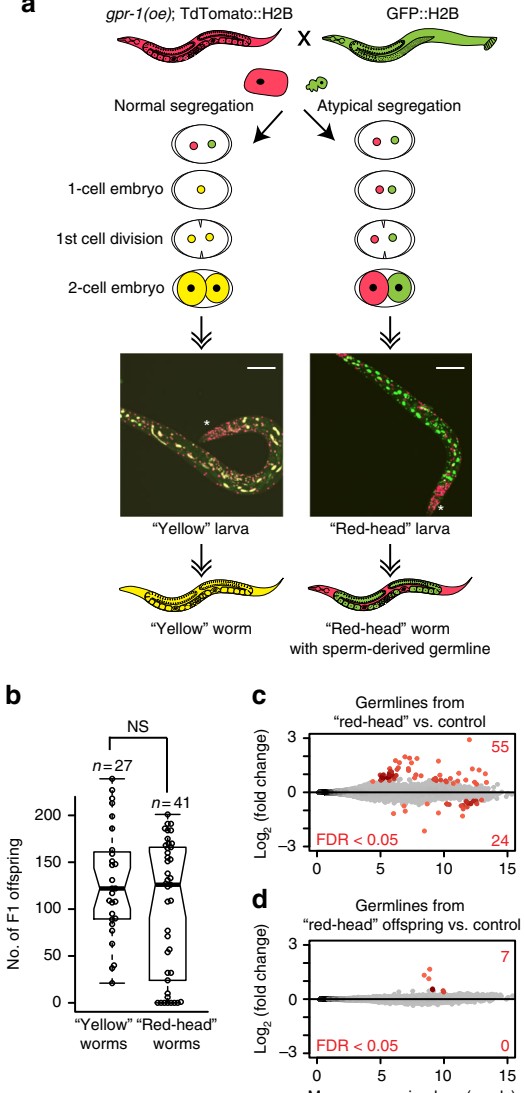

**Fig. 5** The sperm epigenome is sufficient for germline development in offspring. **a** Genetic strategy to test if sperm chromatin marking is sufficient for germ cell development in offspring, using a worm strain that produces some normal-segregation embryos in which each cell inherits both maternal and paternal chromosomes (schematically shown as yellow nuclei) and some atypical-segregation embryos in which the 2 cells of 2-cell embryos inherit either maternal or paternal chromosomes (schematically shown as red and green nuclei). Normal-segregation embryos develop into "yellow" worms. Atypical-segregation embryos develop into "red-head" worms (* marks the head). In "red-head" worms, the germline contains only sperm-derived chromosomes. Scale bars represent 100 μm. **b** Brood size analysis of "yellow" (normal segregation) vs. "red-head" (atypical segregation) F1 offspring, NS (not significant), $p = 0.27$ Mann–Whitney's test. Data from 7 experiments are shown. Each box extends from the 25th to the 75th percentile, with the median indicated by the horizontal line; whiskers extend from the 2.5th to the 97.5th percentile. **c** Differential gene expression analysis (MA plot) of germlines from the "red-head" F1 hermaphrodites (germlines derived solely from sperm chromosomes) vs. GFP::H2B control hermaphrodites (genetically identical control germlines, but derived from both maternal and paternal chromosomes) (top) and their F2 offspring (bottom). Two biological replicates were obtained for each sample. Significantly misregulated genes (FDR < 0.05) were calculated with DESeq2 and are shown in red, and the total numbers of up- and downregulated genes are displayed in the right corners

Intriguingly, multivalent marking of genes with methylated H3K4, H3K36, and H3K27 was also observed in zebrafish sperm[6]. This may be a sperm-specific chromatin signature. Mammalian sperm bear domains of methylated H3K4 and H3K27, similar to bivalent domains observed in embryonic stem cells[3–5,7,35]. In both zebrafish and mammalian sperm, multivalency is associated with genes that are silent during spermatogenesis but expressed during embryo development, suggesting that unique chromatin states in sperm poise certain genes for later expression in embryos. In contrast, multivalency in *C. elegans* sperm is associated with genes that are transcribed during spermatogenesis but are silent in mature sperm and in embryos. We speculate that multivalent marking in *C. elegans* sperm serves a different function than in zebrafish and mammalian sperm, namely to enable an epigenetic memory of spermatogenesis to be reset. Our previous studies suggest that an epigenetic memory of germline gene expression transmitted from sperm and oocytes is maintained by the histone H3K36 methyltransferase MES-4, which is recruited to and maintains high levels of methylated H3K36 on germline-expressed genes in embryos, to help instruct primordial germ cells how to develop[15]. The finding that abnormal retention of an epigenetic memory of spermatogenesis compromises germline health[36] suggests that sperm epigenetic marking must be reset each generation. We speculate that this is achieved in early embryos by removal or loss of active marks (H3K4me3 and H3K36me3) from spermatogenesis-specific genes, to avoid propagating a memory of their expression, and by retaining repressive marks (H3K27me3) on those genes, to keep them repressed until spermatogenesis commences at a later stage. Multivalent marking of chromatin domains may serve diverse functions depending on biological context, developmental tissue and stage, and organism.

Our study demonstrates that *C. elegans* sperm transmit epigenetic information in the form of marked histones, and that this sperm-transmitted information influences development of the next generation. Although mammalian sperm retain a low level of histone packaging, several findings suggest that even mammalian sperm transmit histone-based epigenetic information. First, as noted above, modified histones are retained on developmentally important loci in human and mouse sperm, perhaps to influence expression of those loci in offspring[3,4,7]. Second, single-generation exposure of rats to toxicants resulted in altered patterns of histone retention in sperm in future generations, raising the possibility of transmission of environmental effects in mammals via histone packaging in sperm[37]. Comparisons across organisms continue to shed light on diverse mechanisms of transgenerational epigenetic inheritance and how the life experiences of fathers may impact the development and health of their descendants.

## Methods

***C. elegans* strains and culture conditions**. All strains were cultured on NGM (Nematode Growth Medium) agar plates at 20 °C, unless otherwise noted, using standard methods. CF512 and SS1167 were maintained at 15 °C, and HBR1280 and HBR1281 were maintained at 24 °C. The following strains were used: N2 (Bristol) as wild type, CB1489 *him-8(e1489) IV*, CF512 *rrf-3(b26) II; fem-1(hc17ts) IV*, DH245 *fem-2(b245ts) III*, SS818 *mes-3(bn35) I/hT2-GFP (I;III)*, SS1167 *mes-3 (bn35) I/hT2-GFP (I;III); fem-2(b245ts) III*, HBR1280 *oxTi75[eft-3p::GFP::H2B::tbb-2 3'UTR + unc-18( + )]*, HBR1281 *oxTi411[eft-3p::TdTomato::H2B::unc-54 3'UTR + Cbr-unc-119( + )] III; ddIs32[yfp::gpr-1(synthetic, CAI 1.0, artificial introns)]*.

**Isolation of *C. elegans* males and purification of sperm**. The following protocols were partially adapted from refs. [33,38]. These protocols isolate highly enriched populations of spermatids, which have compacted nuclei that have completed meiotic division but have not developed motility structures to form motile spermatozoa. As these cells have undergone sperm-specific DNA compaction, we refer to these spermatids as mature sperm or "sperm" for short throughout the text. Mature sperm from *C. elegans* were collected and purified from *him-8(e1489)*

males. A mutation in *him-8* causes X chromosome non-disjunction during meiosis and thus generates a high percentage (~ 30%) of males[39]. Large cultures of *him-8 (e1489)* worms (5–10 million) were synchronously grown from starved L1s in fernbach flasks containing S medium and HB101 *Escherichia coli*, shaking at 270 r. p.m. at 20 °C for ~ 62 h (MaxQ 3000 Benchtop shaker, Barnstead Lab-Line). The mixture of gravid hermaphrodites and adult males (day 1 adults) was collected using a separatory funnel. An equal volume of worm suspension and 60% ice-cold sucrose was mixed in a 25 mL tube and centrifuged for 5 min at $1000 \times g$. Worms contained in the top-layer were collected on 20-micron Nytex mesh (CellMicro-Sieves, BioDesign, Inc.) and rinsed with S-basal, after which males were allowed to wiggle through the mesh for at least for 30 min; gravid hermaphrodites remained on top of the mesh. The males were transferred to a 15-micron mesh, rinsed with S-basal several times to remove any larvae and small debris, and collected from the top of the mesh. Filtering of males was repeated twice to yield 10–15 mL packed males of > 98% purity as determined under a dissecting microscope. Males were resuspended in 20 mL freshly prepared Squash Medium (50 mM HEPES, 1 mM MgSO$_4$, 70 mM choline chloride, 5 mM CaCl$_2$, 1 mM phenylmethylsulfonyl fluoride, 1 mg/mL bovine serum albumin, 190 mOsm, pH 6.5), centrifuged for 5 min at $1000 \times g$ and settled on ice, and excess Squash Medium was removed. The packed males were pressed 2 mL at a time using a custom-made "Sperminator" apparatus: males were evenly placed on one Plexiglass "squash plate", a second "squash plate" was placed on top, and the two plates were pressed gently together using the "Sperminator" at 1000 psi for 10 s to squeeze sperm out of male worms, while preserving intact worms to prevent contamination from ruptured worm tissues. The collected sperm were filtered through 20-, 15-, 10-, and two rounds of 5-micron Nytex mesh to eliminate contamination by non-sperm cells and spermatocytes. Purified sperm were collected in a 1.5 mL eppendorf tube by centrifugation ($1700 \times g$ for 5 min), rinsed with Squash Medium, and again filtered through a 5-micron mesh. Sperm quantity (20–100 million sperm) was quantified using a hemocytometer, and sperm purity (~ 99%) was estimated by microscopic analysis (DIC imaging and DNA staining). Pelleted sperm were flash frozen in liquid nitrogen and kept at −80 °C.

**Preparation of sperm chromatin for ChIP**. Frozen sperm (10~50 million) were thawed on ice, rinsed once in 1 mL chilled Micrococcal Nuclease (MNase) Buffer (50 mM HEPES pH 7.5, 110 mM NaCl, 40 mM KCl, 2 mM MgCl$_2$, 1 mM CaCl$_2$), and fixed in 1 mL freshly prepared Fix Solution (1% formaldehyde in MNase Buffer) for 52 s, while constantly pipetting up and down. After adding glycine (final 125 mM) to stop cross-linking, sperm were rinsed once with 1 mL chilled MNase Buffer. Fixed sperm were treated with Lysis Buffer (final 0.2% NP-40 and 0.5% sodium deoxycholate) for 5 min on ice and digested with MNase for 5 min at 37 °C at a density of 0.1 million sperm per µL (5 units of MNase per 1 million sperm). To each tube, EDTA (final 10 mM), protease inhibitor (Roche, #04693159001, final 1 × after a tablet was resuspended in water), and Triton X-100 (final 1%) were added, and each tube was nutated at 4 °C for at least 2 h to facilitate chromatin solubilization. The digested sperm chromatin was mixed with two volumes of 150 mM salt FA Buffer (50 mM of HEPES pH 7.5, 1 mM EDTA pH 7.4, 1% Triton X-100, 0.1% sodium deoxylcholate, 150 mM NaCl) and treated with a Covaris S2 model (duty cycle 20%, intensity 8, cycle per burst 200, 60 s per cycle, total 5 cycles) to liberate sperm chromatin from the insoluble material. The entire content was transferred to a 1.5 mL siliconized tube, and centrifuged at $16,000 \times g$ for 15 min at 4 °C. The supernatant, which contains fragmented and soluble sperm chromatin, was transferred to a new siliconized tube, flash frozen in liquid nitrogen, and stored at −80 °C. Agarose gel electrophoresis (1.2% stained with SYBR Green) and an Agilent 2100 Bioanalyzer System were used to confirm that the majority of soluble chromatin was in mononucleosomes.

**ChIP-seq from sperm**. Soluble sperm chromatin from ~ 3.3 million sperm was thawed on ice, brought up to 110 µL with 150 mM salt FA Buffer supplemented with sarkosyl (final 1%). As "input" 10 µL was saved. To the remaining 100 µL soluble sperm chromatin, 1 µg antibody was added, and ChIP was performed using an IP-Star Compact Automated System (Diagenode) according to the manufacturers' instructions with the following settings and reagents: "indirect method" with 100 µL reaction, 80 µL Dynabeads$^{TM}$ (M-280, sheep anti-mouse IgG, Life Technologies); IP for 12 h, bead incubation for 2 h; 10 min washes at middle speed with 1 M salt FA Buffer, 500 mM salt FA Buffer, TEL Buffer (0.25 M LiCl, 1% NP-40, 1% sodium deoxycholate, 1 mM EDTA, 10 mM Tris-HCl, pH 8), and TE Buffer (10 mM Tris-HCl pH 8.0, 1 mM EDTA); elution in Elution Buffer (1% SDS and 250 mM NaCl in 1X TE). Both IP and "input" were brought up to 300 µL with Elution Buffer with 1.5 µL 20 mg/mL proteinase K and incubated for 2 h at 55 °C and then overnight at 65 °C to reverse crosslinks. The next day, the magnetic beads were removed from the ChIP samples using a magnetic stand and sperm DNA was extracted using phenol–chloroform extraction, using the Phase Lock Gel system (5 PRIME #2302810). Sperm DNA was precipitated overnight at −80 °C in ethanol with glycogen as carrier, and the resulting DNA pellets were resuspended in 15 µL nuclease-free water. One third of the DNA (5 µL) was used for ChIP-quantitative PCR (qPCR) to check ChIP efficiency, and libraries were prepared from the remaining 10 µL ChIP and "input" DNA using MicroPlex Library Preparation kit v2 (Diagenode, #C05010012). A real-time qPCR machine (Roche LightCycler 480) was used to monitor library amplification to avoid

overamplification. Size selection was performed after library amplification using AMPure XP beads (Beckman Coulter, A63881) to enrich for 100–300 bp inserts. The final libraries were evaluated using an Agilent 2100 Bioanalyzer System (High Sensitivity DNA Analysis kit) and Quant-iT assay (Invitrogen, high-sensitivity double-stranded DNA). The multiplexed libraries were sequenced on either HiSeq4000 or HiSeq2500 rapid platforms at the Vincent J. Coates Genomics Sequencing Laboratory at University of California, Berkeley.

**Isolation of *C. elegans* oocytes**. The following protocols were partially adapted from refs. [40–42]. *C. elegans* oocytes were collected and purified from hermaphrodites feminized by temperature-sensitive mutations in *fem-1* and *rrf-3* (previously known as *fer-15*) in strain CF512[43]. At the restrictive temperature, mutant hermaphrodites fail to make sperm and accumulate unfertilized oocytes in the uterus. Synchronized L1 larvae from CF512 were plated on 40–60 High Growth (HG) plates (2% peptone, 51 mM NaCl, 25 mM potassium phosphate, 5 µg/mL cholesterol, 1 mM CaCl$_2$, 1 mM MgCl$_2$, 2.5% agar) seeded with *E. coli* OP50 and grown for ~ 55 h at 15 °C until worms reached late L3 stage. Worms were then upshifted to 25 °C for 24–36 h to feminize them. Visual inspection under a dissecting microscope confirmed that day 1 feminized adults contained only unfertilized oocytes, and that no fertilized embryos or L1 larvae were present on the plates. Approximately 1 million feminized adults were washed from the HG plates with Egg Salts Buffer (25 mM HEPES pH 7.4, 118 mM NaCl, 48 mM KCl, 2 mM CaCl$_2$, 2 mM MgCl$_2$), and washed 3 × in a 50 mL conical tube to remove excess *E. coli* and other debris. After washing, worms were pelleted by centrifugation, excess buffer was removed, and densely packed worms were transferred to a 15 cm petri dish on ice. Worms were chopped with a clean razor blade for 5 min, until extruded gonads and liberated oocytes were visible under a dissecting microscope. Oocytes and carcass fragments were poured over a 45-micron mesh (NTX45, Dynamic Aqua-Supply LTD). Oocytes passed through the mesh, while carcass fragments remained above the mesh. The flow-through was filtered through a 20-micron mesh to collect oocytes above the mesh, while smaller debris went through the mesh. An aliquot of unfixed oocytes was saved for 4′,6-diamidino-2-phenylindole (DAPI) staining and RNA isolation to estimate the quality and purity, measured as the number of oocytes/total cells in multiple fields of view under a dissecting microscope. The purity of the first and second replicates was > 95% and > 90%, respectively. Oocytes were fixed in 2.2% formaldehyde in Egg Salts Buffer for 5 min, quenched in 125 mM glycine to stop cross-linking, and then washed twice in M9 Buffer (3 g KH$_2$PO$_4$, 6 g Na$_2$HPO$_4$, 5 g NaCl, H$_2$O to 1 L). Fixed oocytes were pelleted, flash frozen in liquid nitrogen, and stored at −80 °C.

**ChIP-seq from oocytes**. Fixed oocytes (0.5–1 million) were thawed on ice and resuspended in chilled MNase Buffer. Oocytes were sonicated for 5 cycles with a tip sonicator (5 rounds of 5 s on and 25 s off). Following sonication, oocytes were pre-warmed to 25 °C for 5 min, followed by addition of 250 units MNase. A pilot digestion time-course experiment was performed to identify the optimal digestion time, which yielded the majority of chromatin as mononucleosomes, as measured by Agilent bioanalyzer analysis. As a result of the pilot experiment, oocyte chromatin was digested with MNase for 50 min. Digestion was halted by addition of EDTA (final 10 mM) and incubated on ice for 5 min. Protease inhibitor (Roche, #04693159001, final 1 × in water), Triton X-100 (final 1%), and sodium deoxycholate (final 0.1%) were added to the digested oocyte chromatin extract. The oocyte chromatin extract was split into separate aliquots, and ChIPed in parallel with different antibodies. Dynabeads were washed 3 × with cold FA Buffer. Each antibody (1 µg) was nutated with beads at 4 °C for ~ 2 h. After saving 40 µL oocyte extract as "input," oocyte extract was added to the beads, supplemented with sarkosyl (1% final), and nutated overnight at 4 °C. The next day, the beads were washed twice in FA Buffer, once in 1 M salt FA Buffer, 500 mM salt FA Buffer, and TEL Buffer, and then twice in TE Buffer. Immunocomplexes were eluted from the beads by incubation with Elution Buffer at 65 °C for 15 min with gentle vortexing. RNA was digested with RNase A for 1 h at room temperature, followed by proteinase K treatment for 2 h at 55 °C. Crosslinks were reversed by incubation at 65 °C overnight, followed by phenol–chloroform extraction, and DNA precipitation in ethanol with linear acrylamide at −80 °C for 4 h. ChIPed oocyte DNA was resuspended in 15 µL nuclease-free water. Oocyte libraries were prepared with the NEBNext Ultra DNA library Prep Kit (NEB) following the manufacturers' instructions. AMPure beads were used for size selection after amplification to enrich for fragments corresponding to mononucleosomes. Libraries were sequenced on an Illumina HiSeq2500 platform at the Princeton High Throughput Sequencing Facility.

**ChIP-seq from early embryos**. Data are from ref. [19].

**Antibodies used for ChIP**. Mouse monoclonal antibodies for H3K36me3 (HK00001, now marketed as Wako MABI0333, #300–95289), H3K27me3 (HK00013, now marketed as Wako MABI0323, #309–95259), and H3K4me3 (Wako MABI0304, #305–34819) are described in ref. [44] and were validated as in ref. [45].

**Analysis of ChIP-seq data**. Raw sequence reads from the Illumina HiSeq (50 bp single-end reads) were mapped to the *C. elegans* genome (Wormbase version WS220) using Bowtie with default settings[46]. MACS2 was used to call peaks and to create bedgraph files with the following settings: -g ce --bdg --keep-dup = auto --broad --broad-cutoff = 0.01 --nomodel --extsize = 250[47]. Bedgraph files were scaled to 10 million total autosomal reads (X chromosome reads were not included for scaling, due to the different X:A ratios between sperm, oocytes, and early embryos) and converted to bigwig using the bedGraphToBigWig UCSC Genome Browser tool[48]. These were displayed in the UCSC Genome Browser (visibility = full, autoScale = off, windowingFunction = Mean + whiskers, smoothingWindow = 2). To display gene-level ChIP signal in scatter plots (e.g., Fig. 2c), the mean read coverage for each protein-coding gene was calculated using the bigWigAverageOverBed UCSC Genome Browser tool[48] using WS220 gene start and end annotations. For H3K4me3, a histone mark typically found in promoters, the read coverage was calculated from 500 bp upstream of the gene start annotation to 500 bp downstream. These gene read coverages were log₂ transformed after a pseudo-count of 1 was added. To account for slightly varying background noise levels in the different ChIP-seq samples, the gene read coverages were z-scored based on the average and standard deviation of all autosomal gene read coverages. Specifically, the read coverages for all genes were centered by subtracting the average of the read coverages of autosomal genes and then divided by the standard deviation of the autosomal gene read coverages. One adaptation was made for H3K27me3: the 30th percentile of autosomal gene read coverage, instead of the average, was chosen as the baseline signal and was subtracted from all gene read coverages genome-wide. This is because H3K27me3 occupies ~ 2/3 of the *C. elegans* genome and the autosomal average gene read coverage did not seem like an appropriate baseline to center the H3K27me3 ChIP signal. The Principal Component Analysis in Supplementary Fig. 4d was performed on these z-scored autosomal gene read coverages. The prcomp function in R was used. Metagene analysis was performed using the R package "SeqPlots" with default settings[49].

**MNase-seq**. Sperm were fixed as described for sperm ChIP-seq, and 1–2 million sperm were digested with increasing amounts of MNase for 5 min at 37 °C. Early embryos were harvested from hermaphrodites that contained only a couple of embryos and fixed as described in ref. [15]. Early embryos (500–750 μL) were thawed on ice and resuspended in 2 mL of Dounce Buffer (0.35 M sucrose, 15 mM HEPES pH 7.5, 0.5 mM EGTA, 0.5 mM MgCl₂, 10 mM KCl, 0.1 mM EDTA, 1 mM dithiothreitol, 0.5% Triton X-100) supplemented with protease inhibitors (Roche, #04693159001) and dounce-homogenized ~ 10 times until the majority of embryos were disrupted. The homogenized embryos were spun at $380 \times g$ for 1 min at 4 °C and the supernatant containing nuclei was transferred to a new tube. This was repeated one more time. The nuclei contained in the supernatant were collected by centrifugation at $4000 \times g$ for 10 min at 4 °C and resuspended in MNase Buffer. Approximately 5 million embryo nuclei were digested with increasing amounts of MNase at 37 °C for 30 min. Digested chromatin from sperm and early embryos was reverse-crosslinked overnight at 65 °C in Elution Buffer supplemented with 1.5 μL 20 mg/mL proteinase K. DNA was extracted using phenol–chloroform. The degree of MNase digestion was assessed using agarose gel electrophoresis and an Agilent 2100 Bioanalyzer. Illumina sequencing libraries were constructed with DNA extracted from the mononucleosome-enriched digestions that were comparable between sperm and early embryos, using NEBNext Ultra DNA Library Prep Kit (NEB, E7370). During library construction, size selection was performed on adaptor-ligated DNA before amplification to remove fragments larger than 250 bp (40 μL of AMPure beads were used for First Bead Addition based on the manufacturers' instructions), and PCR-amplified DNA was purified using 1:1 AMPure XP beads, which removed fragments smaller than 100 bp. The final libraries were evaluated using an Agilent 2100 Bioanalyzer System and Quant-iT assay, and were sequenced at the Vincent J. Coates Genomics Sequencing Laboratory at University of California, Berkeley (50 bp paired-end read sequencing).

**Analysis of MNase-seq data**. Paired-end MNase-seq reads were mapped to the *C. elegans* genome (Wormbase version WS220) using Bowtie with default settings[46]. After mapping, fragments smaller than 140 bp were filtered out using bamtools 2.5.1[50]. Bedtools[51] was used to scale the remaining mapped fragments to 10 million autosomal reads and to generate bedgraph files. As for ChIP-seq, the bedGraphToBigWig tool was used to convert the bedgraph files to bigwig format. Bigwig files were displayed in the UCSC genome browser (visibility = full, autoScale = off, windowingFunction = Mean, smoothingWindow = 2). To assess whether there are nucleosome-occupancy differences in certain genomic regions between sperm and early embryos, the bigWigAverageOverBed UCSC Genome Browser tool[48] was used to calculate the average MNase fragment coverage of 150 bp windows tiled every 50 bp and covering the whole genome for all sperm and embryo samples. Windows (150 bp) that had less than 1 fragment covering them in all four sperm and early embryo samples were excluded from further analysis, as these 150 bp windows cover genomic regions of very low mappability with the sequencing data available. About 2.2% of the genome was excluded this way. Density plots were made in R to show the distribution of 150 bp window fragment coverage for the autosomes and for the X chromosome for all four samples of sperm and early embryo MNase-seq. The analysis was repeated for 500 bp windows tiled every 250 bp, 1 kb windows tiled every 500 bp, 2 kb windows tiled every 1 kb, and 5 kb windows tiled every 2.5 kb.

**RNA sequencing**. For male spermatogenic germlines, samples were from N2 males (Figs. 2,3), CB1489 *him-8(e1489)* males (Figs. 2,3,4), and SS818 *mes-3 (bn35)* homozygous M + Z− males from heterozygous parents (M for maternal product and Z for zygotic product) (Fig. 4). For sperm, samples were from CB1489 *him-8 (e1489)* males and SS818 *mes-3(bn35)* M + Z− males (Fig. 4). For hermaphrodite oogenic germlines, samples were from N2 (Figs. 2,3), DH245 *fem-2(b245)* (Figs. 2,3), SS1167 *mes-3(bn35)* (M + P − Z − and M + P + Z− where P is for paternal H3K27me3 by MES-3); *fem-2(b245)* (Fig. 4), HBR1280 *oxTi75[eft-3p::GFP::H2B::tbb-2 3′UTR + unc-18( + )]* (Fig. 5, control), "red-head" F1 progeny (from HBR1281 crossed with HBR1280; genetically identical to HBR1280) (Fig. 5), and F2 offspring from "red-head" F1s (genetically identical to HBR1280) (Fig. 5). Both spermatogenic and oogenic germlines were dissected from day 1 adults as described in ref. [24]; 20–100 distal gonad arms were cut at the late-pachytene gonad bend. Mature sperm were released from adult males by cutting the males with a 30-gauge needle and collected with a pulled glass Pasteur pipette coated with Sigmacote (SL2 Sigma). Total RNA was extracted in TRIzol reagent (Invitrogen), ribosomal RNA was depleted using an NEBNext rRNA Depletion kit (E6310), and libraries were prepared using an NEBNext Ultra RNA Library Prep Kit for Illumina (E7530). Libraries were sequenced at the Vincent J. Coates Genomics Sequencing Laboratory at University of California, Berkeley (50 bp single-end read sequencing). For each genotype, 2–4 biological replicates were obtained. Sequence reads were processed as described in ref. [52]. Briefly, TopHat2[53] was used to align the RNA-seq reads to the *C. elegans* transcriptome (WS220) with default parameters. HTSeq[54] was used to obtain read counts per transcript (HTseq counts). Differentially expressed genes were determined with DESeq2[55] in R using HTSeq counts and a FDR < 0.05 as the significance cutoff. RPKMs were calculated by dividing HTseq counts by exonic transcript length obtained from Wormbase and scaling the total read counts per sample to 1 million reads. If a gene had multiple transcript isoforms, the longest was chosen. For Fig. 3d and Supplementary Fig. 3a,c, RPKMs were log transformed after adding a pseudo-count of 1. Genes with RPKM > 15 were called 'expressed' based on the expression of defined spermatogenic and oogenic gene sets (see below) in spermatogenic and oogenic germlines.

**Single-molecule fluorescence in situ hybridization**. smFISH analysis was performed on day 1 adult male and hermaphrodite gonads dissected from *him-8 (e1489)* worms as in ref. [56]. Biosearch Technologies designed, synthesized, and labeled the Stellaris probes. The *pie-1*, *par-6*, and *nos-2* probes were coupled to Quasar 670, the *mex-3* probe was coupled to Cal Fluor Red 590, and the *cpg-2* probe was coupled to Cal Fluor Red 610. Each probe was resuspended in 250 μL of TE Buffer pH 8.0 and then further diluted to 1:30 for hybridization. The microscope and its settings are as in ref. [52]. Fig. 3 and Supplementary Fig. 6 contain montages generated by splicing together contiguous images acquired with identical settings. Male and hermaphrodite pairs used identical confocal settings, with the exposure optimized for visualizing the male gonads. All images were processed identically with ImageJ and Adobe Illustrator. For both oogenic and male germlines, 10–60 gonads were visually examined under the microscope and at least 3 germlines were imaged in 1–3 separate experiment(s).

**Measurement of fertility/sterility**. For Fig. 4, feminized heterozygous *mes-3 (bn35)/hT2-GFP*; *fem-2ts* worms were obtained by shifting their mothers from 15 °C to 24 °C at the L4 larval stage and selecting GFP + offspring. Feminized worms were mated with either homozygous *mes-3(bn35)* M + Z −; *fem-2ts* males or heterozygous *mes-3(bn35)/hT2-GFP*; *fem-2ts* males raised at 15 °C, and the fertility/sterility of their offspring grown at 15 °C was visually scored under the microscope. We found that having *fem-2ts* and *mes-3* heterozygous in the mother's genotype is essential for parent-effect sterility, and thus represents a sensitized genetic background. Unpaired unequal-variance one-tailed Student's *t*-tests were performed to calculate the significance of % sterile worms between the genetically identical M + P − vs. M + P + offspring.

**Measurement of brood size**. For Fig. 5, strain maintenance and experiments with HBR1280 and HBR1281 were performed at 24 °C, because the transgenes in these strains are prone to gene silencing. Cross-progeny from the matings shown in the figure were scored using a fluorescence microscope to identify those that arose from normal vs. atypical segregation. The numbers of their offspring that grew to adults were scored. We censored 1–3 worms per experiment that died prematurely during the reproductive period (days 1–4 of adulthood) due to the vulva bursting, internal hatching of F1 offspring (bag of worms), or contamination of plates.

**Gene sets**. All gene sets are in Supplementary Data 1.
- Spermatogenesis genes (i.e., spermatogenesis-specific genes): RPKM in spermatogenic germlines (from either wild-type or *him-8* males) > 15, RPKM in wild-type oogenic germlines < 1, and DESeq2 FDR < 0.05 for significance of higher expression in spermatogenic compared to oogenic germlines. Spermatogenic germlines from wild-type males compared to oogenic germlines identified 1369 genes, and *him-8* males compared with oogenic germlines identified 1330 genes.

The 1298 spermatogenesis-specific genes that overlapped between the two comparisons were used in our analyses. The 1298 spermatogenesis genes we defined included 690 of the 827 genes defined in Reinke et al.[21]. The 2498 spermatogenesis genes defined by Ortiz et al.[26] included 1206 of our 1298 genes.

- Oogenesis genes (i.e., oogenesis-enriched genes) are as defined in Reinke et al.[21] (Fig. 3) and Ortiz et al.[26] (Supplementary Fig. 3).

- Sex-independent genes: RPKM in spermatogenic germlines (from either wild-type or *him-8* males) > 5, RPKM in oogenic germlines > 5, FDR > 0.05, and $\log_2$ (fold change) < 2 (genes with no significant differential expression between spermatogenic and oogenic germlines). Spermatogenic germlines from wild-type males compared with oogenic germlines identified 2549 genes, and spermatogenic germlines from *him-8* males compared with oogenic germlines identified 2472 genes. The 2097 overlapping genes from these two comparisons were used in our analyses.

- Soma-specific genes (1171 genes) are genes expressed in at least 1 of 3 somatic tissues (muscle, gut, or neuron) with at least 8 SAGE tags[57] but not enriched[21] or detectably expressed[58] in the adult germline.

- Silent genes are 410 serpentine receptor genes that are expressed in a few mature neurons and are not expected to be expressed in germlines, originally defined in[23].

**Statistics**. Statistical analysis and sample sizes for all experiments are described in figure legends. Student's *t*-test was performed using Excel. Mann–Whitney *U*-test and hypergeometric test were performed using R. At least 2 biological replicates of sequencing data (ChIP/MNase/RNA-seq) were obtained; sample sizes were based on experience and the standards in the field. We omitted sequencing data generated and analyzed during the optimization phase of the project. The final samples included were those generated after optimization. DESeq2 was used to test for differential expression by use of negative binomial generalized linear models. Adjusted *p*-values were calculated by DESeq2 using the Benjamini–Hochberg method.

**Data availability**. Raw sequence reads and processed data generated or reanalyzed in this study (MNase-seq, ChIP-seq, and RNA-seq) are in the NCBI GEO database, accession number GSE115709. All gene sets (see above) are in Supplementary Data 1. Other data used in this study are available at the NCBI GEO database, accession numbers GSE715-GSE737 (ref. [21]), GSE57109 (ref. [26]), ArrayExpress: E-TABM-598 (ref. [23]), http://elegans.bcgsc.ca/home/sage.html (ref. [57]), and http://tock.bcgsc.ca/cgi-bin/sage160 (ref. [58]).

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

## Acknowledgements

We thank Strome lab members and A. Fire for helpful discussions, K. Kaneshiro for oogenic germline transcriptome data, E. Nishimura for some smFISH probes, H. Kimura for antibodies, D. Chu and J. Lieb for advice, and H. Bringmann for sharing *gpr-1(oe)* strains prior to publication. Some strains were provided by the CGC funded by NIH-P40OD010440. This work was supported by NIH-F32GM110892 to T.M.T., NIH-T32HG003284 to T.E.J., NIH-DP1GM119167 to C.T.M., and NIH-R01GM34059 to S.S., and used the QB3 Vincent J. Coates Genomics Sequencing Laboratory at UC Berkeley funded by S10OD018174.

## Author contributions

T.M.T. and S.S. designed the study. T.M.T., T.E.J., and T.A.E. performed the experiments. T.E.J. and C.T.M. provided the oocyte data. T.M.T. and A.R. performed bioinformatic analysis. T.M.T., A.R., and S.S. wrote the paper with help from T.A.E.

## Additional information

**Competing interests:** The authors declare no competing interests.

