## [Peer Review File · Nature Communications]

Reviewers' comments:

Reviewer #1 (Remarks to the Author):

Using the nematode *C. elegans*, the authors of the submitted manuscript have performed several histone modification profiles by ChIP-seq (namely H3K27me3, H3K36me3, and H3K4me3) in chromatin extracts from sperm and oocytes. These profiles have never been obtained in oocytes and mature sperm of *C. elegans*, and therefore this work constitutes a useful resource of data. Interestingly, the sperm epigenome is decorated with H3K27me3, H3K36me3, and H3K4me3 marks. The profiles signature is similar to the one previously generated in early embryo preparation, and to the oocyte epigenome. This is in contrast with the common idea that the sperm epigenome is highly compacted with non-histone proteins, such as protamines. The analyses of ChIP-seq and RNA-seq data also show interesting features of the sperm epigenome. First, genes involved in spermatogenesis are marked with either active and silent marks (trivalent marks). Second, genes that are normally expressed during oogenesis are also expressed, at some extent, in the male germline and are marked in the sperm epigenome with active chromatin modifications. Finally, they demonstrated that H3K27me3 inherited from sperm is required for fertility, and that germlines derived from cells that are constituted exclusively of sperm epigenome developed normally. Based on these results, they concluded that paternal epigenetic transmission is sufficient for germline development. However, this work does not go further to study 1) the nature and the role of the trivalent marks on spermatogenesis genes, 2) the transcriptional changes that determine the sterility phenotype in worms derived from sperm devoid of H3K27me3, 3) the role of oogenesis gene transcripts in the male germline.

Overall, I think that this work is suitable to be published in Nature Communication. Nevertheless, I do have some major comments that the author should address before publication. For instance, the levels of each chromatin modifications detected in sperm chromatin relative to other cell types (oocytes or embryos) have not been evaluated in this study. My concern is that the levels of histone modifications detected in sperm chromatin might be much lower than embryos or oocytes chromatin, at global level or on specific set of genes. This concern comes from the visual inspection of their figures showing snapshots of the ChIP-seq profiles across the genome (Fig 1a,b, S1b, S3). Because the authors haven't used the same scale (y-axes) between different samples for a given histone modification, the levels of some profiles in the sperm might be even ten time less than the embryo profile. For this reason, a proper relative quantification of each epigenetic chromatin modifications between sperm, oocytes and embryos across the genome or gene body is required. Moreover, although the authors claim that every region of the sperm genome is covered by nucleosome, their analysis of MNase-seq does not provide many details and in the present state is barely convincing. Finally, the authors should add much more information about the methodology used, especially for the part related to the data analysis and normalization.

Below my specific comments to be addressed:

1. The genome-wide view of the whole genome by MNase-seq presented in Figure S1a is not very informative. As a matter of fact, a more detailed analysis of the size of the fragments they have obtained after paired-end sequencing could reveal canonical nucleosome (around 150bp) and subnucleosomal particle (less than 80bp). It is not clear from the manuscript (and in the method section) whether they have conduct such analysis (select only fragments of a certain size for their subsequent analysis) or they have just mapped on the genome all the sequences, regardless to their size distribution. If they have done such type of data manipulation, they can analyze more precisely only the fragments corresponding to the canonical nucleosome size and exclude other types of MNase protection (they should comment on this). Also, a deeper analysis of their MNase-seq may show differences at the promoters and gene body of specific classes of genes (such as spermatogenesis and oogenesis genes) in the sperm and embryo genome. Furthermore, they can computationally identify regions of the sperm genome covered by other types of proteins or non-canonical nucleosome.

2. The authors should consider to quantify the signal of each mark on the gene bodies in sperm vs embryos, oocytes vs embryos, sperm vs Oocytes. The proper quantification of the chromatin marks in sperm compared to the Oocyte and Embryo will help to understand the relative amount of these modifications. At the present, the data only shows a qualitative view on the epigenetic landscape of the sperm chromatin. In addition, the authors should use the same scale for all the figures containing snapshot of the genome with histone methylation profile. They should pay attention to set up a scaling factor using the highest peak. In many figures (especially in the embryo dataset), the majority of peaks go over the threshold imposed (y-axis scale). Moreover, in figure 1a the scale of ChIP-seq of the histone modifications is quite different between sperm and embryo. Notably, the sperm H3K36me3 signal is more than 10 times lower than the H3K36me3 signal in embryo. If the profiles shown correspond to the normalized read counts (please specify in the legend), then this data might suggest that there is much lower methylated histone in the sperm epigenome.

3. The level of expression of spermatogenesis genes has been measured in male germline and not mature sperm (where the ChIP-seq profile has been generated), which is usually transcriptionally silenced. Therefore, the active chromatin marks (H3K36me3 and H3K4me3) 1) might be maintained in absence of transcription, 2) might be transcriptionally active although at very low level, 3) might be the consequences of the composition of their sperm preparation, which might contain a percentage of sperm from meiotic stages or earlier stages. In the first two cases the authors should check whether in their sperm preparation spermatogenesis genes are devoid of transcription (using smFISH, Pol II ChIP, etc.). In the third case the signal from active and silent chromatin marks might come from sperm in different developmental stages. Therefore, the authors should show that their preparation of sperm is composed of mainly mature spermatozoa.

4. The oocytes used in this study has been collected from a mutant (*fem-1*) that accumulate unfertilized oocytes. However, it has been shown that oocytes dissected from *fem-1* mutant have thousands of differentially expressed genes compared to the oocytes from manually dissected WT worms (Stoeckius et al., 2014, PMID: 24957527). My concern is that the chromatin and the RNA extracted from *fem-1* mutant oocytes might show profiles that are different from wild type oocytes. Thus, I suggest that the authors at least compare the RNA-seq from Stoeckius et al. with their ChIP-seq and RNA-seq.

5. In figure 4a, the authors show that offspring from male mutant for *mes-3* becomes sterile. However, it would be useful to know which category of genes are mis-regulated in these sterile animals and whether *mes-3* mutation cause some up-regulation and accumulation of transcripts in mature sperm that are then transmitted to the progeny. In order to identify which category of genes is affected in these sterile animals, the authors should perform RT-qPCR (or smFISH) on selected oogenesis genes, spermatogenesis genes, and sex-independent genes (unless is possible to perform RNA-seq). Same kind of assays should be performed in germlines and mature sperm derived from *mes-3* mutant male.

6. In the supplementary method section, the authors should describe better the kind of analysis they have performed for the MNase-seq data. Also, for the ChIP-seq data they should explain better the kind of normalization adopted. For example, is not clear to me why they scaled the data so that "the variance of the autosomal gene read averages was 1". Furthermore, the kind of normalization they have used to show the genomic view of their data is not clear.

Minor comments:

1. In figure 2b the input signal from sperm chromatin on spermatogenesis genes is not constant and shows variation at the TSS. The authors should comment on this.

2. In supplementary figure 3, they forgot to highlight in blue the spermatogenesis gene (next to *pie-1*). Moreover, the chromatin profiles of this gene from this replica is quite different from figure 3a. The H3K36me3 and H3K4me3 is almost absent for this spermatogenesis gene (and is similar to the silent gene) compared to the replica in figure 3a. Also, on the left panel the gene C14A4.8 and C14A4.13 shows low levels of H3K36me3 compared to figure 2a. These differences might indicate that the two biological replicates might differ in level of methylation in some important regions. This should be taken into account. Perhaps a PCA analysis of the two replicates would

help. Finally, the gene annotation on the top of the figure is different from the main figures 2a,b 3a (genes shows as arrows).

3. Four hypotheses have been proposed for the presence of trivalent marks. However, only one of them has been addressed experimentally. Therefore, I propose that the authors either remove the figure S4 or try to address the four possibilities. Again, another possible explanation might be that they detect the different chromatin marks from mature sperm in different stages of development.

4. The authors have only tested the role of inherited H3K27me3 from sperm using *mes-3* mutant, but they don't use mutants for H3K36me3 and H3k4me3. Thus, I am wondering whether these experiments are feasible and can be performed.

5. On page 3 the authors write "We note that protamine-like proteins have been identified in *C. elegans*, and it is possible that some regions are packaged with histones in some sperm and with protamines in other sperm". It is not clear to me what the authors mean for "some sperm" and "other sperm".

Reviewer #2 (Remarks to the Author):

This manuscript describes the marking of paternal chromatin with histone post-translational modifications and the influence of those markings before and after fertilization on transcription and development. Given the interest in epigenetics on development and fertility, the work fills an important gap in knowledge about the role of paternal influence on these processes. Overall the work is impactful and well done. I recommend it highly for publication.

I have two moderate concerns. First, the authors state that sperm carry a memory of oogenesis gene expression. This gives the impression that sperm remember the transcriptional status during oogenesis. This should be reframed to more accurately reflect that sperm express genes that have been previously found to have enriched expression during oogenesis. Second, the authors suggest the term "trivalency" for the multiple histone marking observed in sperm. Two issues with this term: 1) Multivalency is a term used by other researchers – why not use the same term? 2) The description of how the gene body marking compares to bivalency or multivalency observed in different organisms is lacking. There should be a more in depth discussion about how the multi-marking observed in *C. elegans* compares to that seen in other organisms. Overall, I suspect the authors' brevity is a result of word count limits - the manuscript is already very lean and well-written. I would advocate for leeway on the word count limit because many of the issues that are raised in the detailed comments below can and must be addressed by more thorough descriptions of important aspects of the work and more discussion of the findings and implications.

Detailed comments:

Abstract: "Here we report that *Caenorhabditis elegans* sperm carry a histone-based epigenetic memory of spermatogenesis gene expression, and surprisingly of oogenesis gene expression as well."

The 'oogenesis gene expression' as a description of these genes does not accurately reflect that the authors show these genes are actually expressed during spermatogenesis. By phrasing it this way, the authors reinforce that these genes are only expressed during oogenesis. The fact that they are expressed during spermatogenesis too is actually an interesting and surprising finding they could better highlight.

Page 3 Ln 3 – It may be relevant to also note that *C. elegans* lacks DNA methylation, thus may rely more on histone post-translational modifications to transmit epigenetic information.

Page 3 "We found no nucleosome-devoid regions of the genome in sperm or early embryos." At the resolution shown in the Supplementary Figure 1a, this is difficult to evaluate that NO regions are nucleosome free. It is hard to imagine nucleosome occupancy doesn't vary at all between

sperm and embryo given the different transcriptional differences between cell types...is this what the authors mean? What do they really define as a 'region'?

Because the authors show both sperm and early embryo data, it may be more relevant to describe these findings based on a comparison of the two. This is not meant to require additional full analyses to define differences – it is meant to instead ensure the level of occupancy that the authors describe are accurate when they say NO regions are nucleosome free and that ALL regions are the genome are packaged.

Page 4 Ln 1 "H3K36me3 and H3K27me3 generally occupying mutually exclusive domains across each autosome, similar to those observed in early embryos." It may be helpful to have more specific terms than "generally occupying mutually exclusive domains". What is a domain? Later the authors will say there are 'categories' of genes that break this rule, which is difficult to reconcile if in the previous paragraphs it has repeatedly been said that the localization of these marks are mutually exclusive. It is important that the authors distinguish and define what they mean by the terms region, domain, and category.

Page 4 "Comparison of the distribution of H3K36me3 and H3K27me3 in sperm, oocytes, and early embryos revealed that all 3 stages display the typical mutually exclusive patterns of those marks over most regions of the genome."

This sentence is hard to understand. Does this mean that each doesn't overlap with the other two? Or that active marks don't overlap with repressive marks? In general, this paragraph has a lot of information and needs a more clear topic sentence to better frame the main point.

Page 4 "To display genome-wide marking of genes..."

The term "genome-wide marking of genes" is confusing. Not sure what it means.

In Figure 2C and D, it would be helpful to remove the "all gene" in gray from the yellow and the blue graphs to better visualize the extent of enrichment of the sex-independent and spermatogenesis categories. The "All genes" is included in the left panel anyways.

Page 5 "In short, both germline-expressed and germline-silent genes in both spermatogenic and oogenic germlines follow the 'either H3K36me3 or H3K27me3' rule, consistent with the known antagonism between H3K36me3 and H3K27me." This sentence is difficult to follow.

Page 5 Ln "These are genes expressed exclusively in spermatogenic germlines and not in oogenic germlines."

The term "exclusive" is a bit misleading. Both of the cited studies defined these as "enriched." Using the term 'exclusive' makes it a bit hard to follow why the authors find the oogenesis genes expressed during spermatogenesis, while the term 'enriched' more accurately reflects that they could be expressed, just not as abundantly.

Page 5 A suggestion is to consider referring to the marking of genes with H3K36me3, H3K4me3, and H3K27me3 as 'multivalent' instead of 'trivalent'. The cited zebrafish study refers to the 'similar' marking as 'multivalent', not trivalent. Also, it may be that other histone modification or features will be found in the future to correlate with these marks. The term 'multivalent' will work if so but trivalent will not.

Page 5 – is the localization of H3K36me and H3K27me3 over the gene bodies of spermatogenesis genes unique to *C. elegans* spermatogenesis? Are genes that have been shown to have bivalency or multivalency over gene bodies or promoters?

Page 6 "These are genes expressed exclusively during oogenesis, which would be expected to bear H3K27me3 in sperm."

Same issue with the term “exclusively” as described above.

Page 7 “To test the latter, we analyzed our germline RNA-seq data and found that 73% of ‘oogenesis genes’ (82% of those bearing the active mark H3K36me3 in sperm) are indeed transcribed (RPKM>15) in spermatogenic germlines”

What does the RPKM value represent? How does that value relate to genes that are highly expressed in sperm? Or of these genes expressed in oocytes or embryos? It would be important to know that relative expression level in relation to other genes or other cell types so that one could get an idea of the transcription level of ‘oogenesis’ genes during spermatogenesis.

Page 7 “We conclude that sperm carry an epigenetic memory of both spermatogenesis and oogenesis gene expression in the form of histone modifications.”

It seems odd to say that sperm carry a memory of oogenesis gene expression. These genes, though important for oocyte function and expressed in oocytes, were actually expressed during spermatogenesis. Thus, sperm are carrying an epigenetic memory of spermatogenesis gene expression.

Reviewer #3 (Remarks to the Author):

In this study, Tabuchi et al. examine the epigenetic contributions of sperm and oocyte to the *C. elegans* embryo. They show that, like many organisms and as been previously observed in zebrafish, that *C. elegans* sperm genome is packed with nucleosomes, and that the histones in the nucleosomes carry histone methylation patterns that are largely consistent with gene expression patterns during embryogenesis: genes that are not expressed during spermatogenesis or oogenesis are enriched for the repressive modification, H3K27 methylation, and those expressed are marked by H3K36 and H3K4 methylation, modifications established by transcription. Surprisingly, genes with spermatogenesis-restricted expression are marked by all three modifications, and this pattern is resolved to these genes being solely marked by H3K37me3 in embryos. This indicates that sperm-expressed genes are subjected to reprogramming, which has been suggested by other studies—at least for H3K4 methylation-- in *C. elegans* (e.g., Katz et al., 2009). The authors also show that “oogenic” genes are expressed during spermatogenesis and are thus marked by H3K36me3 and H3K4me3 but lack H3K27me3 in sperm chromatin. This is not necessarily surprising, given the prominence of post-transcriptional regulation of germ cell sex determination in this hermaphroditic species. The authors further investigate whether H3K27me3 marking of sperm chromatin is necessary and sufficient for germline development. They show that sperm lacking H3K27me3 cannot support germ cell development in the offspring produced by these sperm. They further test sufficiency using a very clever trick to produce offspring whose germ lineage only inherit paternal genomes, and these animals are indeed fertile. This is not necessarily surprising, given that (as they authors appropriately reference) Hodgkin genetically demonstrated an absence of parental imprinting in *C. elegans* over 25 years ago. However, this study does make a stronger case for both equal genetic and epigenetic contributions by both parents in this organism.

Overall the study expands the current concept that the paternal epigenome can make a significant contribution to the offspring, an idea that has been somewhat suppressed by the large-scale remodeling of mammalian sperm chromatin—a situation that is clearly not the norm in nature. What the study lacks somewhat is evidence that any transgenerational effects that have been reported in worms can be observed as disruptions in the normal patterns that they observe in this study—although this would significantly extend the scope of this work. Yet, the work as is remains largely observational—i.e., they observe an “unusual combination of histone modifications” but

little is offered by way of ideas of how or why.

Other Comments:

The study leaves an impression that H3K27me3 may be the main modification that is required to maintain fertility. Is this true? It is important to at least discuss their previous data on H3K36me3 and any existing data on H3K4me3's role in this epigenetic "memory" system, and their ideas regarding whether they are simply consequences of transcription or also contribute mechanistically.

How does sperm chromatin accumulate H3K27me3 in actively transcribing genes? Does the MES Polycomb system only come onto these genes after transcription ceases? Can this be readily observed in their system?

The authors describe a cohort of "oogenesis genes", yet many of these genes are not involved in oogenesis per se, but are maternally synthesized factors important for early embryonic development. It might be useful to many readers to clarify this to avoid confusion of interpretation of which types of genes are being analyzed.

Minor Comments:

It is not clear why they also used a "healthier version of a GPR-1 overexpressing mutant" or what it adds to the study—is there something we should be concerned about with the "less healthy" overexpressing mutant? The reason I ask is that this requires the addition of a "personal communications" reference, which some journals don't permit.

Figure 3A: The Germline transcripts" and male/herm labels are mis-aligned with the data

We thank the reviewers for their many good suggestions for improving our manuscript. We have addressed all concerns, which included performing additional experiments and new analyses. And we reformatted the manuscript to Nature Communication format, which allowed expanded discussion. Our responses to the reviewers' suggestions comments are in blue italics below. Our major changes to the manuscript are highlighted in yellow in the resubmission and include:

- Deeper analysis of our MNase-seq data from sperm versus early embryos (new Supplementary Fig. 2b).
- PCA analysis of ChIP-seq samples and replicates (new Supplementary Fig. 4d).
- RNA-seq analysis of *mes-3* male parent germlines, their sperm, and the germlines of their M+P- offspring (all with controls) to investigate whether an altered payload of RNAs in sperm leads to altered transcription in the germlines of offspring (new Fig. 4c,d).
- Providing a table of all gene sets that we analyzed (new Supplementary Data 1).

Reviewer #1:

Using the nematode *C. elegans*, the authors of the submitted manuscript have performed several histone modification profiles by ChIP-seq (namely H3K27me3, H3K36me3, and H3K4me3) in chromatin extracts from sperm and oocytes. These profiles have never been obtained in oocytes and mature sperm of *C. elegans*, and therefore this work constitutes a useful resource of data. Interestingly, the sperm epigenome is decorated with H3K27me3, H3K36me3, and H3K4me3 marks. The profiles signature is similar to the one previously generated in early embryo preparation, and to the oocyte epigenome. This is in contrast with the common idea that the sperm epigenome is highly compacted with non-histone proteins, such as protamines.

The analyses of ChIP-seq and RNA-seq data also show interesting features of the sperm epigenome. First, genes involved in spermatogenesis are marked with either active and silent marks (trivalent marks). Second, genes that are normally expressed during oogenesis are also expressed, at some extent, in the male germline and are marked in the sperm epigenome with active chromatin modifications. Finally, they demonstrated that H3K27me3 inherited from sperm is required for fertility, and that germlines derived from cells that are constituted exclusively of sperm epigenome developed normally. Based on these results, they concluded that paternal epigenetic transmission is sufficient for germline development. However, this work does not go further to study 1) the nature and the role of the trivalent marks on spermatogenesis genes, 2) the transcriptional changes that determine the sterility phenotype in worms derived from sperm devoid of H3K27me3, 3) the role of oogenesis gene transcripts in the male germline.

Overall, I think that this work is suitable to be published in Nature Communication. Nevertheless, I do have some major comments that the author should address before publication.

For instance, the levels of each chromatin modifications detected in sperm chromatin relative to other cell types (oocytes or embryos) have not been evaluated in this study. My concern is that the levels of histone modifications detected in sperm chromatin might be much lower than embryos or oocytes chromatin, at global level or on specific set of genes. This concern comes from the visual inspection of their figures showing snapshots of the ChIP-seq profiles across the genome (Fig 1a,b, S1b, S3). Because the authors haven't used the same scale (y-axes) between different samples for a given histone modification, the levels of some profiles in the sperm might be even ten time less than the embryo profile. For this reason, a proper relative quantification of each epigenetic chromatin modifications between sperm, oocytes and embryos across the genome or gene body is required.

We now show normalized ChIP-seq reads using the same scales for sperm, oocytes, and early embryos. We normalized all ChIP-seq reads to 10 million autosomal reads. We did not include X chromosome reads for normalization, because sperm have a different X:A ratio than oocytes and early embryos. For genome-browser snapshots, the y-axis maximum values are 360 for H3K36me3, 220 for H3K27me3, and 1200 for H3K4me3 for all samples throughout the manuscript. H3K36me3 and H3K27me3 show similar distributions and peak sizes between sperm and early embryos. This suggests that the levels of H3K36me3- and H3K27me3-modified nucleosomes are comparable between sperm and early embryos. H3K4me3 is present at reduced levels in sperm compared to early embryos.

Moreover, although the authors claim that every region of the sperm genome is covered by nucleosome, their analysis of MNase-seq does not provide many details and in the present state is barely convincing.

Please see our response to specific comment 1 below.

Finally, the authors should add much more information about the methodology used, especially for the part related to the data analysis and normalization.

We provided more information in Methods.

Below my specific comments to be addressed:

1. The genome-wide view of the whole genome by MNase-seq presented in Figure S1a is not very informative. As a matter of fact, a more detailed analysis of the size of the fragments they have obtained after paired-end sequencing could reveal canonical nucleosome (around 150bp) and subnucleosomal particle (less than 80bp). It is not clear from the manuscript (and in the method section) whether they have conducted such analysis (select only fragments of a certain size for their subsequent analysis) or they have just mapped on the genome all the sequences, regardless to their size distribution. If they have done such type of data manipulation, they can analyze more precisely only the fragments corresponding to the canonical nucleosome size and exclude other types of MNase protection (they should comment on this). Also, a deeper analysis of their MNase-seq may show differences at the promoters and gene body of specific classes of genes (such as spermatogenesis and oogenesis genes) in the sperm and embryo genome. Furthermore, they can computationally identify regions of the sperm genome covered by other types of proteins or non-canonical nucleosome.

As requested by Reviewers 1 and 2, we performed deeper analysis of our MNase-seq data to determine occupancy of canonical nucleosomes in sperm and early embryos. We had size-selected MNase fragments (100-250 bp) during library preparation, so we could not analyze fragments of <80 bp. To focus on canonical nucleosomes, we analyzed fragments of >140 bp.

First, we observed no visible nucleosome-free domains at a chromosome scale in sperm compared with early embryos (Supplementary Fig. 2a). Second, to compare nucleosome occupancy between the two in detail, we analyzed MNase-seq reads over various window sizes for autosomes and the X chromosome, and displayed them as density plots (Supplementary Fig. 2b). The plots show that MNase-seq reads from sperm and early embryos are superimposable for autosomes, regardless of window size, suggesting that nucleosome occupancy between sperm and early embryos is highly comparable. For the X chromosome, average sequence reads for sperm (half have an X chromosome, half lack an X chromosome) were shifted to the left of early embryos by ~2-fold, as expected. We conclude that the sperm genome retains nucleosomes genome-wide, similar to early embryos. Third, as requested, we performed metagene analysis of our MNase-seq data to investigate nucleosome occupancy at promoter regions of various classes of genes (Fig. 1 for Reviewers, provided at the end of this document). The most pronounced nucleosome-depleted region is at the promoters of spermatogenesis genes in sperm compared to early embryos and compared to the promoters of other categories of genes in sperm.

2. The authors should consider to quantify the signal of each mark on the gene bodies in sperm vs embryos, oocytes vs embryos, sperm vs Oocytes. The proper quantification of the chromatin marks in sperm compared to the Oocyte and Embryo will help to understand the

relative amount of these modifications. At the present, the data only shows a qualitative view on the epigenetic landscape of the sperm chromatin.

As described above, ChIP-seq reads were normalized to 10 million autosomal reads, and metagene analysis was performed with normalized reads, instead of z-scores (in the original figure), over various categories of genes (Fig. 2b). We found that the patterns and the levels of H3K36me3 and H3K27me3 over sex-independent and silent genes are similar among sperm, oocytes, and embryos, suggesting that these three samples have comparable distributions and levels of H3K36me3- and H3K27me3-marked nucleosomes. This analysis also showed that the level of H3K4me3 is lower in oocytes and sperm than early embryos, as seen in mouse studies^{1,2}.

In addition, the authors should use the same scale for all the figures containing snapshot of the genome with histone methylation profile. They should pay attention to set up a scaling factor using the highest peak. In many figures (especially in the embryo dataset), the majority of peaks go over the threshold imposed (y-axis scale). Moreover, in figure 1a the scale of ChIP-seq of the histone modifications is quite different between sperm and embryo. Notably, the sperm H3K36me3 signal is more than 10 times lower than the H3K36me3 signal in embryo. If the profiles shown correspond to the normalized read counts (please specify in the legend), then this data might suggest that there is much lower methylated histone in the sperm epigenome.

See our response above.

3. The level of expression of spermatogenesis genes has been measured in male germline and not mature sperm (where the ChIP-seq profile has been generated), which is usually transcriptionally silenced. Therefore, the active chromatin marks (H3K36me3 and K3K4me3) 1) might be maintained in absence of transcription, 2) might be transcriptionally active although at very low level, 3) might be the consequences of the composition of their sperm preparation, which might contain a percentage of sperm from meiotic stages or earlier stages. In the first two cases the authors should check whether in their sperm preparation spermatogenesis genes are devoid of transcription (using smFISH, Pol II ChIP, etc.). In the third case the signal from active and silent chromatin marks might come from sperm in different developmental stages. Therefore, the authors should show that their preparation of sperm is composed of mainly mature spermatozoa.

To address the third point, using our optimized sperm purification protocol, we estimated the purity to be ~99% mature sperm free of spermatocytes and other types of cells, based on visual inspection (Methods). We added a representative image of purified sperm (and oocytes) used for ChIP experiments in Supplementary Fig. 1.

Regarding the second point, it has been shown that, unlike flies and mammals, C. elegans spermatogenic germlines lack a 'post-meiotic burst' of transcription; instead transcription ceases during chromosome condensation prior to the sperm meiotic divisions and remains repressed in post-meiotic spermatids^{3,4}. This was demonstrated by immunostaining of spermatogenic germlines, which showed no detectable active RNA polymerase II (pCTD-ser2) or acetylated histones in spermatocytes during and after sperm meiosis.

Taken together, our data align best with the reviewer's first point that active histone marks (H3K36me3 and H3K4me3) are maintained in sperm in the absence of transcription. Indeed, previous studies have shown that both H3K36me and H3K4me can be maintained in the absence of ongoing transcription^{5,6}.

4. The oocytes used in this study has been collected from a mutant (fem-1) that accumulate unfertilized oocytes. However, it has been shown that oocytes dissected from fem-1 mutant have thousands of differentially expressed genes compared to the oocytes from manually dissected WT worms (Stoeckius et al., 2014, PMID: 24957527). My concern is that the chromatin and the RNA extracted from fem-1 mutant oocytes might show profiles that are different from wild type oocytes. Thus, I suggest that the authors at least compare the RNA-seq from Stoeckius et al. with their ChIP-seq and RNA-seq.

The use of ovulated oocytes from feminized animals ('feminized oocytes' for short) was unavoidable for this study, as it is currently not possible to purify enough wild-type oocytes to perform ChIP-seq. While we acknowledge the limitation of using ovulated oocytes from feminized worms on page 4 of the manuscript, our detailed analyses below suggest that those oocytes reflect the chromatin state of wild-type oocytes prior to fertilization, and that that state is closely related to the state in early embryos.

Our comparison of the transcriptomes of the feminized oocytes used in our study versus hand-picked wild-type oocytes⁷ indeed revealed numerous changes in transcript levels; in particular down-regulation of germline genes and up-regulation of somatic genes in feminized oocytes suggest that feminized oocytes are prematurely heading into embryogenesis. When we compared ChIP-seq data from feminized oocytes with ChIP-seq data from wild-type early embryos, we found very strong correlations ($R^2=0.90$ for H3K36me3, and 0.79 for H3K27me3), suggesting that the chromatin states of the two are very similar (Fig. 2 for Reviewers, provided at the end of this document). When genes significantly down-regulated (germline-enriched genes - blue) and up-regulated (somatic genes - pink) in feminized oocytes vs. wild-type oocytes are highlighted in the same plots, both sets of genes fall onto the diagonal line, indicating little change in H3K36me3 and H3K27me3 for these genes in feminized oocytes vs. wild-type early embryos. Notably, down-regulated germline-enriched genes are marked with active H3K36me3 and lack

repressive H3K27me3 in both oocytes and embryos, and up-regulated somatic genes are marked with repressive H3K27me3 and lack active H3K36me3. These findings suggest that changes in gene expression in feminized oocytes have negligible effects on H3K36me3- or H3K27me3-marked chromatin, and are consistent with a model proposed by Rechtsteiner et al.⁵ that H3K36me3 and H3K27me3 transmit an epigenetic memory of germline gene expression and repression and do not reflect ongoing transcription in early embryos.

We include some of the above discussion in the main text (page 4, line 10). If the reviewers think our ChIP-seq data from feminized oocytes are misleading, the oocyte data can be removed from the paper with little impact on the overall story, since the focus of our paper is on sperm vs. early embryo chromatin states.

In case the reviewers wonder, we are currently switching to a new technique, CUT&RUN^{8,9}, to profile chromatin states from small amounts of material (e.g. wild-type unfertilized oocytes), but we do not yet have it working.

5. In figure 4a, the authors show that offspring from male mutant for *mes-3* becomes sterile. However, it would be useful to know which category of genes are mis-regulated in these sterile animals and whether *mes-3* mutation cause some up-regulation and accumulation of transcripts in mature sperm that are then transmitted to the progeny. In order to identify which category of genes is affected in these sterile animals, the authors should perform RT-qPCR (or smFISH) on selected oogenesis genes, spermatogenesis genes, and sex-independent genes (unless is possible to perform RNA-seq). Same kind of assays should be performed in germlines and mature sperm derived from *mes-3* mutant male.

*To address the reviewer's interesting idea that *mes-3* sperm may transmit an mRNA memory of gene expression changes to offspring, we performed RNA-seq from dissected male germlines (*mes-3* vs. control males), from their mature sperm (*mes-3* vs. control sperm), and from dissected germlines from F1 offspring (M+P- vs. M+P+ hermaphrodites). The results, shown in Fig. 4c,d, indicate that differentially sperm-packaged transcripts do not predict gene mis-regulation in the germlines of M+P-offspring, arguing that sperm do not transmit an mRNA-based memory. Instead, differentially expressed genes in male parent germlines predict differentially expressed genes in the germlines of M+P- offspring, consistent with a chromatin-based memory. We found that genes up-regulated in the germlines of M+P- offspring are enriched for somatic genes and X-linked genes, which may underlie the sterility of M+P- offspring. Somatic and X-linked genes are up-regulated in M+P+Z- *mes* mutants¹⁰; we did not further investigate them in this paper.*

6. In the supplementary method section, the authors should describe better the kind of analysis they have performed for the MNase-seq data. Also, for the ChIP-seq data they should explain better the kind of normalization adopted. For example, is not clear to me why

they scaled the data so that “the variance of the autosomal gene read averages was 1”. Furthermore, the kind of normalization they have used to show the genomic view of their data is not clear.

We provided more detailed descriptions of MNase-seq, ChIP-seq, and normalization in Methods.

Minor comments:

1. In figure 2b the input signal from sperm chromatin on spermatogenesis genes is not constant and shows variation at the TSS. The authors should comment on this.

All input profiles in Fig. 2b are fairly flat after ChIP-seq reads were normalized to 10 million autosomal reads, and metagene analysis was performed with normalized reads, instead of z-scores (in the original figure), which can amplify small variations. Our added MNase-seq analysis (Fig. 1 for Reviewers) shows that in sperm chromatin, spermatogenesis gene promoters are nucleosome-depleted. Consistent with this, our MNase-treated sperm Input shows a slight dip in signal in the promoter region of spermatogenesis genes as well.

2. In supplementary figure 3, they forgot to highlight in blue the spermatogenesis gene (next to pie-1). Moreover, the chromatin profiles of this gene from this replica is quite different from figure 3a. The H3K36me3 and H3K4me3 is almost absent for this spermatogenesis gene (and is similar to the silent gene) compared to the replica in figure 3a. Also, on the left panel the gene C14A4.8 and C14A4.13 shows low levels of H3K36me3 compared to figure 2a. These differences might indicate that the two biological replicates might differ in level of methylation in some important regions. This should be taken into account. Perhaps a PCA analysis of the two replicates would help. Finally, the gene annotation on the top of the figure is different from the main figures 2a,b 3a (genes shows as arrows).

We thank the reviewer for catching our Supplementary Fig. 3 mistake and suggesting PCA analysis. Our file conversion for Supplementary Fig. 3 accidentally merged the panels showing 2 oogenic loci (pie-1 and par-6). In our new supplementary figure (now Supplementary Fig. 4), we show 8 boxed genes in 3 panels for replicate 2. The tracks for replicates 1 and 2 look very similar. We also added to Supplementary Fig. 4d PCA analysis of all ChIP-seq data.

3. Four hypotheses have been proposed for the presence of trivalent marks. However, only one of them has been addressed experimentally. Therefore, I propose that the authors either

remove the figure S4 or try to address the four possibilities. Again, another possible explanation might be that they detect the different chromatin marks from mature sperm in different stages of development.

As suggested, we removed the original Supplementary Fig. 4. As noted above, our preparations of mature sperm were 99% pure, so the additional possibility mentioned above (different chromatin marks from sperm in different stages of development) is highly unlikely.

4. The authors have only tested the role of inherited H3K27me3 from sperm using *mes-3* mutant, but they don't use mutants for H3K36me3 and H3k4me3. Thus, I am wondering whether these experiments are feasible and can be performed.

*Loss of PRC2 activity in *mes-3* mutants causes levels of H3K27me3 to drop below detection in both spermatogenic and oogenic germlines, making it feasible to test the impact of loss of that mark. In contrast, H3K36me3 and H3K4me3 are each generated by more than 1 histone methyltransferase (HMT). Loss of H3K36me3 requires loss of 2 H3K36me3 HMTs (MES-4 and MET-1), and *mes-4; met-1* double mutant strains are hard to maintain and work with (J. Kreher and S. Strome, manuscript in preparation), making our experiments infeasible. Similarly, multiple H3K4me3 HMTs must be knocked out to eliminate H3K4me⁶. Furthermore, our analysis of histone marking in early embryos suggests that inherited patterns of H3K4me3 are quickly converted to transcription-coupled patterns of H3K4me3 (unpublished), making the inherited pattern less likely to serve an important role in offspring. We added to the main text some discussion of these points on page 7 line 1.*

5. On page 3 the authors write "We note that protamine-like proteins have been identified in *C. elegans*, and it is possible that some regions are packaged with histones in some sperm and with protamines in other sperm". It is not clear to me what the authors mean for "some sperm" and "other sperm".

*We modified the main text on page 9 line 8 to "It is possible that in *C. elegans* sperm some regions are packaged with a mixture of histones and protamines, as has been proposed in human sperm".*

Reviewer #2:

This manuscript describes the marking of paternal chromatin with histone post-translational modifications and the influence of those markings before and after fertilization on transcription and development. Given the interest in epigenetics on development and fertility,

the work fills an important gap in knowledge about the role of paternal influence on these processes. Overall the work is impactful and well done. I recommend it highly for publication.

I have two moderate concerns. First, the authors state that sperm carry a memory of oogenesis gene expression. This gives the impression that sperm remember the transcriptional status during oogenesis. This should be reframed to more accurately reflect that sperm express genes that have been previously found to have enriched expression during oogenesis. Second, the authors suggest the term “trivalency” for the multiple histone marking observed in sperm. Two issues with this term: 1) Multivalency is a term used by other researchers – why not use the same term? 2) The description of how the gene body marking compares to bivalency or multivalency observed in different organisms is lacking. There should be a more in depth discussion about how the multi-marking observed in *C. elegans* compares to that seen in other organisms. Overall, I suspect the authors’ brevity is a result of word count limits - the manuscript is already very lean and well-written. I would advocate for leeway on the word count limit because many of the issues that are raised in the detailed comments below can and must be addressed by more thorough descriptions of important aspects of the work and more discussion of the findings and implications.

As suggested by the reviewer, we clarified the definition of oogenesis genes in the main text on page 5 line 26 as “genes with oogenesis-enriched expression (‘oogenesis-enriched genes’ or ‘oogenesis genes’ for short, Methods). Some gene products from this class are directly involved in oogenesis (e.g. the yolk receptor RME-2), while others are maternally synthesized and oocyte-supplied gene products involved in early embryo development (e.g. EGG-4, PIE-1, PAR-6, and NOS-2)”.

*We also changed the term “trivalency” to “multivalency” throughout the manuscript, and added a paragraph to Discussion (starting on page 9 line 18) comparing multivalency in *C. elegans* sperm to multivalency observed in other organisms and discussing possible roles of multivalency.*

Detailed comments:

Abstract: “Here we report that *Caenorhabditis elegans* sperm carry a histone-based epigenetic memory of spermatogenesis gene expression, and surprisingly of oogenesis gene expression as well.”

The ‘oogenesis gene expression’ as a description of these genes does not accurately reflect that the authors show these genes are actually expressed during spermatogenesis. By phrasing it this way, the authors reinforce that these genes are only expressed during oogenesis. The fact that they are expressed during spermatogenesis too is actually an interesting and surprising finding they could better highlight.

We rephrased the text about oogenesis gene expression, as noted above.

Page 3 Ln 3 – It may be relevant to also note that *C. elegans* lacks DNA methylation, thus may rely more on histone post-translational modifications to transmit epigenetic information.

We added the suggested sentence to the main text on page 2 line 20: “C. elegans offers an exceptional system in which to address these questions. C. elegans lacks DNA methylation on cytosines, one established mediator of epigenetic control, and thus may rely more heavily on histone modifications to transmit epigenetic information”.

Page 3 “We found no nucleosome-devoid regions of the genome in sperm or early embryos.” At the resolution shown in the Supplementary Figure 1a, this is difficult to evaluate that NO regions are nucleosome free. It is hard to imagine nucleosome occupancy doesn't vary at all between sperm and embryo given the different transcriptional differences between cell types...is this what the authors mean? What do they really define as a 'region'? Because the authors show both sperm and early embryo data, it may be more relevant to describe these findings based on a comparison of the two. This is not meant to require additional full analyses to define differences – it is meant to instead ensure the level of occupancy that the authors describe are accurate when they say NO regions are nucleosome free and that ALL regions of the genome are packaged.

Please see our response to Reviewer 1 specific comment 1, describing our more thorough analysis of the MNase-seq data, which is presented in new Supplementary Fig. 2.

Page 4 Ln 1 “H3K36me3 and H3K27me3 generally occupying mutually exclusive domains across each autosome, similar to those observed in early embryos.” It may be helpful to have more specific terms than “generally occupying mutually exclusive domains”. What is a domain? Later the authors will say there are 'categories' of genes that break this rule, which is difficult to reconcile if in the previous paragraphs it has repeatedly been said that the localization of these marks are mutually exclusive. It is important that the authors distinguish and define what they mean by the terms region, domain, and category.

We modified the text to better describe “domains” on page 3 line 26: “We found that C. elegans sperm retain modified histones across all six chromosomes with alternating H3K36me3- and H3K27me3-marked chromatin domains across the five autosomes (Fig. 1, Supplementary Fig. 2c). This domain organization is similar to that observed in early embryos: H3K36me3 domains span the coding regions of single genes or sets of adjacent genes, and H3K27me3 domains overlie silent genes and intergenic regions”.

Page 4 “Comparison of the distribution of H3K36me3 and H3K27me3 in sperm, oocytes, and early embryos revealed that all 3 stages display the typical mutually exclusive patterns of those marks over most regions of the genome.”

This sentence is hard to understand. Does this mean that each doesn't overlap with the other two? Or that active marks don't overlap with repressive marks? In general, this paragraph has a lot of information and needs a more clear topic sentence to better frame the main point.

We modified the text as requested on page 4 line 21: “Comparison of ChIP-seq data revealed that sperm, oocytes, and early embryos display very similar chromatin domains marked with either H3K36me3 or H3K27me3”.

Page 4 “To display genome-wide marking of genes...”

The term “genome-wide marking of genes” is confusing. Not sure what it means.

We modified the text as requested on page 4 line 24: “To ask whether genes are marked with H3K36me3, H3K27me3, both, or neither ...”.

In Figure 2C and D, it would be helpful to remove the “all gene” in gray from the yellow and the blue graphs to better visualize the extent of enrichment of the sex-independent and spermatogenesis categories. The “All genes” is included in the left panel anyways.

We added the suggested separate-color panels as Supplementary Fig. 5.

Page 5 “In short, both germline-expressed and germline-silent genes in both spermatogenic and oogenic germlines follow the ‘either H3K36me3 or H3K27me3’ rule, consistent with the known antagonism between H3K36me3 and H3K27me.” This sentence is difficult to follow.

We modified the text as requested on page 5 line 6: “In short, genes expressed in both spermatogenic and oogenic germlines are marked with H3K36me3, and genes silent in both are marked with H3K27me3, following an ‘either H3K36me3 or H3K27me3’ rule, consistent with the known antagonism between H3K36me3 and H3K27me3”.

Page 5 Ln “These are genes expressed exclusively in spermatogenic germlines and not in oogenic germlines.”

The term “exclusive” is a bit misleading. Both of the cited studies defined these as “enriched.”

Using the term ‘exclusive’ makes it a bit hard to follow why the authors find the oogenesis genes expressed during spermatogenesis, while the term ‘enriched’ more accurately reflects that they could be expressed, just not as abundantly.

In this study, we defined “spermatogenesis-specific genes” in Methods as those that are expressed in spermatogenic germlines and NOT in oogenic germlines. We also highlight “spermatogenesis-enriched genes” defined by Reinke et al. (2003)¹¹ and Ortiz et al. (2014)¹² in Supplementary Fig. 3 to compare to our spermatogenesis-specific genes. To clarify these points, the main text was edited and now reads on page 5 line 11: “These are genes expressed exclusively in spermatogenic germlines and not in oogenic germlines (‘spermatogenesis-specific genes’ or ‘spermatogenesis genes’ for short, Supplementary Fig. 3a, Methods). Strikingly, these genes (as well as previously defined sperm-enriched genes) display both H3K36me3 and H3K27me3 over their gene body in sperm”.

Page 5 A suggestion is to consider referring to the marking of genes with H3K36me3, H3K4me3, and H3K27me3 as ‘multivalent’ instead of ‘trivalent’. The cited zebrafish study refers to the ‘similar’ marking as ‘multivalent’, not trivalent. Also, it may be that other histone modification or features will be found in the future to correlate with these marks. The term ‘multivalent’ will work if so but trivalent will not.

We made this wording change throughout the paper, as noted above.

Page 5 – is the localization of H3K36me and H3K27me3 over the gene bodies of spermatogenesis genes unique to *C. elegans* spermatogenesis? Are genes that have been shown to have bivalency or multivalency over gene bodies or promoters?

The colocalization of H3K36me3 and H3K27me3 is reported in zebrafish sperm as being over gene promoters and in some cases extending over gene bodies of single or adjacent developmental genes, not spermatogenesis genes¹³. These developmental genes are also marked with H3K4me3 (multivalency). We added a new paragraph on multivalency to Discussion, as noted above.

Page 6 “These are genes expressed exclusively during oogenesis, which would be expected to bear H3K27me3 in sperm.”

Same issue with the term “exclusively” as described above.

Please see our reply above.

Page 7 “To test the latter, we analyzed our germline RNA-seq data and found that 73% of 'oogenesis genes' (82% of those bearing the active mark H3K36me3 in sperm) are indeed transcribed (RPKM>15) in spermatogenic germlines”

What does the RPKM value represent? How does that value relate to genes that are highly expressed in sperm? Or of these genes expressed in oocytes or embryos? It would be important to know that relative expression level in relation to other genes or other cell types so that one could get an idea of the transcription level of 'oogenesis' genes during spermatogenesis.

In this new submission, we defined genes with RPKM >15 as 'expressed'. RPKM (Reads Per Kilobase of transcript, per Million mapped reads) values represent normalized expression levels of genes. We plotted the distribution of RPKM for all coding genes in density plots (Fig. 3 for Reviewers, provided at the end of this document). All genes (black line) in both male and oogenic germlines show two peaks: a strong peak below RPKM of 1 (silent genes) and a broad peak above RPKM 15 (expressed genes). In male germlines, 28% of all genes have RPKM >15. In oogenic germlines, 25% of all genes have RPKM values >15. Also in Fig. 3 for Reviewers, spermatogenesis-specific genes are shown in blue, and oogenesis genes are shown in red. In male germlines, spermatogenesis genes peak around RPKM = 100 ($\log_{10}(100) = 2$) and oogenesis genes peak around RPKM = 30 ($\log_{10}(30) = 1.48$). This shows that, in male germlines, oogenesis genes are indeed expressed at least as well as the average 'expressed genes', but less robustly than spermatogenesis genes. In oogenic germlines, oogenesis genes peak around RPKM = 60 ($\log_{10}(60) = 1.78$) and spermatogenesis genes peak below RPKM of 1. This suggests that oogenesis genes are more robustly expressed (about 2-fold more) in oogenic germlines compared to male germlines.

To clarify these points, we added lines to Fig. 3d and Supplementary Fig. 3a,c to mark RPKM = 15 ($\log_{10}(15) = 1.2$) to show the relative expression of 'oogenesis genes' in spermatogenic vs. oogenic germlines. We also expanded the description of RNA-seq in Methods.

Page 7 “We conclude that sperm carry an epigenetic memory of both spermatogenesis and oogenesis gene expression in the form of histone modifications.”

It seems odd to say that sperm carry a memory of oogenesis gene expression. These genes, though important for oocyte function and expressed in oocytes, were actually expressed during spermatogenesis. Thus, sperm are carrying an epigenetic memory of spermatogenesis gene expression.

We modified the main text on page 6 line 22: “We conclude that sperm carry a histone-based epigenetic memory of spermatogenesis gene expression, which includes genes well known for their expression during oogenesis.”

Reviewer #3:

In this study, Tabuchi et al. examine the epigenetic contributions of sperm and oocyte to the *C. elegans* embryo. They show that, like many organisms and as been previously observed in zebrafish, that *C. elegans* sperm genome is packed with nucleosomes, and that the histones in the nucleosomes carry histone methylation patterns that are largely consistent with gene expression patterns during embryogenesis: genes that are not expressed during spermatogenesis or oogenesis are enriched for the repressive modification, H3K27 methylation, and those expressed are marked by H3K36 and H3K4 methylation, modifications established by transcription. Surprisingly, genes with spermatogenesis-restricted expression are marked by all three modifications, and this pattern is resolved to these genes being solely marked by H3K37me3 in embryos. This indicates that sperm-expressed genes are subjected to reprogramming, which has been suggested by other studies—at least for H3K4 methylation-- in *C.elegans* (e.g., Katz et al., 2009).

Indeed, our analysis of the epigenetic landscapes of sperm, oocytes, and early embryos at the individual gene level reveals surprising multivalent marking of spermatogenesis-restricted genes and loss of both H3K4me3 and H3K36me3 from those genes in embryos, complementing and extending the genetic study of Katz et al. (2009)¹⁴.

The authors also show that “oogenic” genes are expressed during spermatogenesis and are thus marked by H3K36me3 and H3K4me3 but lack H3K27me3 in sperm chromatin. This is not necessarily surprising, given the prominence of post-transcriptional regulation of germ cell sex determination in this hermaphroditic species.

To our knowledge, this is the first genomic study that demonstrates that genes that were thought to be devoted to oogenesis and provisioning early embryos are transcribed in spermatogenic germlines from males and accordingly marked with active histone modifications in sperm.

The authors further investigate whether H3K27me3 marking of sperm chromatin is necessary and sufficient for germline development. They show that sperm lacking H3K27me3 cannot support germ cell development in the offspring produced by these sperm. They further test sufficiency using a very clever trick to produce offspring whose germ lineage only inherit paternal genomes, and these animals are indeed fertile. This is not necessarily surprising, given that (as they authors appropriately reference) Hodgkin genetically demonstrated an absence of parental imprinting in *C. elegans* over 25 years ago. However, this study does make a stronger case for both equal genetic and epigenetic contributions by both parents in this organism.

*Our findings are indeed consistent with those of Haack and Hodgkin (1991)¹⁵ that showed that offspring that inherit both homologs of a marked autosome from the oocyte or the sperm are viable and fertile, indicating that *C. elegans* autosomes lack imprinting. Here we demonstrate that the entire genome, not individual pairs of autosomes, inherited from the sperm is sufficient to support normal germline development. Furthermore, given the importance of sperm epigenetic marking, we infer that the sperm epigenome is also sufficient.*

Overall the study expands the current concept that the paternal epigenome can make a significant contribution to the offspring, an idea that has been somewhat suppressed by the large-scale remodeling of mammalian sperm chromatin—a situation that is clearly not the norm in nature. What the study lacks somewhat is evidence that any transgenerational effects that have been reported in worms can be observed as disruptions in the normal patterns that they observe in this study—although this would significantly extend the scope of this work. Yet, the work as is remains largely observational—i.e., they observe an “unusual combination of histone modifications” but little is offered by way of ideas of how or why.

The molecular nature and function(s) of multivalent marking of spermatogenesis genes in sperm are of great interest to us but require involved analyses that are beyond the scope of the current paper. For example, we are trying to develop a nanopore-based approach to investigate whether marks are on the same nucleosomes, and we will use mass spectrometry to investigate whether marks are on the same H3 tails. As noted above, we added a new paragraph on multivalency to Discussion.

Other Comments:

The study leaves an impression that H3K27me3 may be the main modification that is required to maintain fertility. Is this true? It is important to at least discuss their previous data on H3K36me3 and any existing data on H3K4me3's role in this epigenetic "memory" system, and their ideas regarding whether they are simply consequences of transcription or also contribute mechanistically.

Please see our response to Reviewer 1 minor comment 4.

How does sperm chromatin accumulate H3K27me3 in actively transcribing genes? Does the MES Polycomb system only come onto these genes after transcription ceases? Can this be readily observed in their system?

We first would like to determine if H3K36me3 and H3K27me3 co-occur on the same nucleosomes by nanopore, re-ChIP, or a microscope-based method¹⁶⁻¹⁹. If they do co-occur, two scenarios below are possible.

First, as the reviewer suggested, PRC2 may be recruited to H3K36me3-marked spermatogenesis genes after transcription ceases. We think this is unlikely, because pre-existing H3K36 methylation antagonizes PRC2 activity in vitro²⁰. Furthermore, this scenario would suggest that all actively transcribed genes (including sex-independent genes) should acquire H3K27me3 along with H3K36me3, but spermatogenesis genes are unique in acquiring both marks.

*Another possibility is that pre-existing H3K27me3-marked spermatogenesis genes (perhaps in primordial germ cells) allow recruitment of an H3K36 HMT during spermatogenesis-specific transcription. This is plausible given that preexisting H3K27me3 does not inhibit H3K36 methylation²⁰. Previous ChIP-chip/seq analyses of *C. elegans* early embryos and L3 larvae detected no H3K36me3 over spermatogenesis genes^{21,22}. To further test this possibility, we are adapting an alternative to ChIP-seq (called CUT&RUN, discussed above) to profile histone marks in primordial germ cells and in dissected germlines from males (spermatogenic) vs females (oogenic) and at different stages of germ cell development.*

The authors describe a cohort of "oogenesis genes", yet many of these genes are not involved in oogenesis per se, but are maternally synthesized factors important for early embryonic development. It might be useful to many readers to clarify this to avoid confusion of interpretation of which types of genes are being analyzed.

We modified the text accordingly on page 5 line 26, as noted above.

Minor Comments:

It is not clear why they also used a “healthier version of a GPR-1 overexpressing mutant” or what it adds to the study—is there something we should be concerned about with the “less healthy” overexpressing mutant? The reason I ask is that this requires the addition of a “personal communications” reference, which some journals don’t permit.

The original GPR-1(OE) strain was difficult to handle because multi-copy transgenes often undergo silencing. Thus, we repeated the experiment with a “healthier version of a GPR-1 overexpressing mutant” created by CRISPR, which does not undergo silencing. We removed the supplementary figure that used the healthier version.

Figure 3A: The Germline transcripts” and male/herm labels are mis-aligned with the data

The misalignment of labels has been fixed.

Figure 1 for Reviewers. Spermatogenesis genes display a pronounced nucleosome-depleted region at their promoters. Normalized MNase-seq signals around (1 kb up- and

Figure 2 for Reviewers. ChIP-seq data from fem-1 oocytes vs. wild-type early embryos show high correlations. The mean normalized H3K36me3 (top row) or H3K27me3 (bottom row) ChIP signals for all coding genes (grey) in fem-1 oocytes vs. wild-type early embryos, highlighting the significantly up-regulated somatic genes in fem-1 oocytes compared to wild-type oocytes (pink) and the significantly down-regulated germline-enriched genes in fem-1 oocytes compared to wild-type oocytes (blue). The

Figure 3 for Reviewers. In male germlines, oogenesis genes are expressed close to the average for suppressed genes, but less robustly than

1. Zhang, B. *et al.* Allelic reprogramming of the histone modification H3K4me3 in early mammalian development. *Nature* **537**, 553–557 (2016).
2. Dahl, J. A. *et al.* Broad histone H3K4me3 domains in mouse oocytes modulate maternal-to-zygotic transition. *Nature* **537**, 548–552 (2016).
3. Shakes, D. C. *et al.* Spermatogenesis-specific features of the meiotic program in *Caenorhabditis elegans*. *PLoS Genet.* **5**, 1000611 (2009).
4. Samson, M. *et al.* The specification and global reprogramming of histone epigenetic marks during gamete formation and early embryo development in *C. elegans*. *PLoS Genet.* **10**, 17–21 (2014).
5. Rechtsteiner, A. *et al.* The histone H3K36 methyltransferase MES-4 acts epigenetically to transmit the memory of germline gene expression to progeny. *PLoS Genet* **6**, e1001091 (2010).
6. Li, T. & Kelly, W. G. A role for Set1 / MLL-related components in epigenetic regulation of the *Caenorhabditis elegans* germ line. *PLoS Genet.* **7**, e1001349 (2011).
7. Stoeckius, M. *et al.* Global characterization of the oocyte-to-embryo transition in *Caenorhabditis elegans* uncovers a novel mRNA clearance mechanism. *EMBO J.* **33**, 1751–1766 (2014).
8. Skene, P. J. & Henikoff, S. An efficient targeted nuclease strategy for high-resolution mapping of DNA binding sites. *Elife* **6**, 1–35 (2017).
9. Skene, P. J., Henikoff, J. G. & Henikoff, S. Targeted in situ genome-wide profiling with high efficiency for low cell numbers. *Nat. Protoc.* **13**, 1006–1019 (2018).
10. Gaydos, L. J., Rechtsteiner, A., Egelhofer, T. A., Carroll, C. R. & Strome, S. Antagonism between MES-4 and Polycomb Repressive Complex 2 promotes appropriate gene expression in *C. elegans* germ cells. *Cell Rep* **2**, 1169–1177 (2012).
11. Reinke, V., Gil, I. S., Ward, S. & Kazmer, K. Genome-wide germline-enriched and sex-biased expression profiles in *Caenorhabditis elegans*. *Development* **131**, 311–323 (2004).
12. Ortiz, M. A., Noble, D., Sorokin, E. P. & Kimble, J. A new dataset of spermatogenic vs. oogenic transcriptomes in the nematode *Caenorhabditis elegans*. *G3* **4**, 1765–1772 (2014).
13. Wu, S. F., Zhang, H. & Cairns, B. R. Genes for embryo development are packaged in blocks of multivalent chromatin in zebrafish sperm. *Genome Res* **21**, 578–589 (2011).
14. Katz, D. J., Edwards, T. M., Reinke, V. & Kelly, W. G. A *C. elegans* LSD1 demethylase contributes to germline immortality by reprogramming epigenetic memory. *Cell* **137**, 308–320 (2009).
15. Haack, H. & Hodgkin, J. Tests for parental imprinting in the nematode *Caenorhabditis*

- elegans*. *MGG Mol. Gen. Genet.* **228**, 482–485 (1991).
16. Weiner, A. *et al.* Co-ChIP enables genome-wide mapping of histone mark co-occurrence at single-molecule resolution. *Nat. Biotechnol.* **34**, 953–961 (2016).
 17. Kinkley, S. *et al.* ReChIP-seq reveals widespread bivalency of H3K4me3 and H3K27me3 in CD4+memory T cells. *Nat. Commun.* **7**, (2016).
 18. Sadeh, R., Launer-Wachs, R., Wandel, H., Rahat, A. & Friedman, N. Elucidating combinatorial chromatin states at single-nucleosome resolution. *Mol. Cell* **63**, 1079–1088 (2016).
 19. Shema, E. *et al.* Single-molecule decoding of combinatorially modified nucleosomes. *Science.* **352**, 717–721 (2016).
 20. Yuan, W. *et al.* H3K36 methylation antagonizes PRC2-mediated H3K27 methylation. *J Biol Chem* **286**, 7983–7989 (2011).
 21. Liu, T. *et al.* Broad chromosomal domains of histone modification patterns in *C. elegans*. *Genome Res* **21**, 227–236 (2011).
 22. Ho, J. W. K. *et al.* Comparative analysis of metazoan chromatin organization. *Nature* **512**, 449–452 (2014).

REVIEWERS' COMMENTS:

Reviewer #1 (Remarks to the Author):

I thank the authors for addressing all my previous concerns. In particular they have performed a more in depth analysis of their MNase-seq, a better quantification of each epigenetic chromatin modifications analysed by ChIP-seq, and an RNA-seq of mes-3 mutant offspring. Therefore, I recommend their manuscript for publication.

Reviewer #2 (Remarks to the Author):

The revised manuscript by Tabuchi et al describes epigenetic contributions from sperm and characterizes the fate and importance of these paternal contributions to the embryo. The work is a significant contribution to our knowledge of the function and dynamics of epigenetic information that is passed from generation to generation. It uses elegant experiments to show that paternal epigenetic information is critical for germline development in the offspring. Both the quality, quantity, and scope of work are impressive. The revision has significantly addressed the major concerns raised in the initial review. As such, I recommend it highly for publication in Nature Communications.

There are no major issues with the data or interpretations. Listed below are specific comments and suggestions to clarify the description of the work.

Specific Comments:

1) One issue that re-occurs is the description of 'oogenesis genes.' The authors show that genes whose expression is enriched during oogenesis are actually expressed during spermatogenesis. This is a surprising and, as they point out in the results section (Pg 5 Ln 137), 'unexpected.' The authors need to point out more consistently that this was not expected based on other studies (or explain about the nature of the 'enrichment' in oogenesis). In particular:

-Abstract Pg 1 Ln 21: "Here we report that in *Caenorhabditis elegans* sperm, the genome is packaged in nucleosomes and carries a histone-based epigenetic memory of genes expressed during spermatogenesis, which includes genes well known for their expression during oogenesis."

Change to (suggested wording): "Here we report that in *Caenorhabditis elegans* sperm, the genome is packaged in nucleosomes and carries a histone-based epigenetic memory of genes expressed during spermatogenesis, which unexpectedly includes genes well known for their expression during oogenesis."

-Introduction Pg 3 Ln 59: "Oogenesis-enriched genes are marked in sperm with active marks as a result of their transcription during spermatogenesis."

Change to (suggested wording): "We also found genes previously found to have enriched expression during oogenesis are also transcribed during spermatogenesis and thus bear active marks as a result of their transcription."

2) There is another confusion about the oogenesis gene class. On Pg 5 Ln 11 They define the sex-independent class as those genes that are "expressed in both spermatogenic and oogenic germlines - these genes have "the active modification H3K36me3 and lack the repressive modification H3K27me3."

However, later, the authors find that many "oogenesis genes" are actually transcribed in spermatogenesis (Pg 6 Ln147). However, these genes bear "H3K36me3, and 75% with H3K4me3, and are devoid of H3K27me3"

This brings up the following questions:

- It is not clear from the data cited (Fig. 3d, Supplementary Fig. 3c, Methods) how this statement is true: "we analyzed our germline RNA-seq data and found that 73% of oogenesis genes (82% of those bearing the active mark H3K36me3 in sperm) are indeed transcribed in spermatogenic germlines." For example, what subset in Fig 3d is the 73%? Is there some information about the enrichment level of expression that would help clarify this?
- Since "oogenesis genes" are actually transcribed in sperm, are they in the sex-independent class? Are the oogenesis genes and sex-independent genes distinct or overlapping classes? It is unclear how the 'oogenesis' class relates to some of the other classes defined because those other classes were defined by this study, while the oogenesis class seems to be defined only by other studies (Pg 20 Ln 539). Did the authors find oogenesis-specific genes that have different markings? This is not meant to ask for further analyses or experiments– it is asking for clarification about the oogenesis gene class and how it is differentiated from the sex-independent gene class.
- Are the markings of oogenesis genes and sex-independent genes distinct? Do the sex-independent class of genes also have H3K4me3?

3) Pg 3 Ln 72: "We found that sperm retain nucleosomes across all six chromosomes, comparable to early embryos.."

To clarify 'comparable' because there is both presence or gross distribution (which I think the authors mean) and levels, which is what is referred to about X in the latter half of the sentence. Are both the levels and distribution comparable? In Supplemental Figure 2b: The 'density' is not well explained – a brief explanation would be helpful to more quickly clarify these issues for the reader – density, coverage, distribution, and levels should be clarified.

4) Pg 6 Ln 161: "We conclude that sperm carry a histone-based epigenetic memory of spermatogenesis gene expression, which includes genes well known for their expression during oogenesis."

This conclusion does not point out that this 'epigenetic memory' marking is distinct – that these 'oogenesis genes' carry only active marks and not the repressive H3K27me3 mark. This is in contrast to the spermatogenesis-genes that have the multivalent mark. To be clear, it would be helpful in this conclusion point to be specific that the marking is different but each still constitutes a memory of gene expression during spermatogenesis.

5) Pg 7 Ln 181: Because MES-3 has been shown to be important for silencing X chromosomes in the hermaphrodite germline, can the authors note whether or not the genes that show differential expression in the mes-3 mutants are X genes?

6) A general note: The authors use "sperm" throughout the manuscript. The cells isolated by the methods in this manuscript are technically unactivated spermatids that have not developed motility apparatuses to form active spermatozoa. An issue with the use of this term is that sperm used in studies from other organisms are usually spermatozoa. It is highly unlikely that this affects the results and conclusions of this work because in *C. elegans*, sperm-specific chromatin remodeling occurs concurrent to meiotic divisions and before activation. Nonetheless, the distinction between the cell types should be explained so that other researchers are aware of the differences of *C. elegans* and vertebrate sperm biology. To do this, include in the methods section (Pg 11 Ln 284 after "adapted from.") something like "These protocols isolate highly enriched populations of spermatids, which have compacted nuclei that have completed meiotic division but have not developed motility structures to form motile spermatozoa. Because these cells have undergone sperm-specific DNA compaction, we refer to these spermatids as mature sperm throughout the text."

7) Supplemental Figure 1: add scale bars

Reviewer #3 (Remarks to the Author):

All of my concerns about the previous submission have been addressed in this revised submission.

We thank Reviewers #1 and #3 for approving our resubmission and Reviewer #2 for the excellent suggested wording changes. Our responses are in blue italics below.

Reviewer #1 (Remarks to the Author):

I thank the authors for addressing all my previous concerns. In particular they have performed a more in depth analysis of their MNase-seq, a better quantification of each epigenetic chromatin modifications analysed by ChIP-seq, and an RNA-seq of mes-3 mutant offspring. Therefore, I recommend their manuscript for publication.

Reviewer #2 (Remarks to the Author):

The revised manuscript by Tabuchi et al describes epigenetic contributions from sperm and characterizes the fate and importance of these paternal contributions to the embryo. The work is a significant contribution to our knowledge of the function and dynamics of epigenetic information that is passed from generation to generation. It uses elegant experiments to show that paternal epigenetic information is critical for germline development in the offspring. Both the quality, quantity, and scope of work are impressive. The revision has significantly addressed the major concerns raised in the initial review. As such, I recommend it highly for publication in Nature Communications.

There are no major issues with the data or interpretations. Listed below are specific comments and suggestions to clarify the description of the work.

Specific Comments:

1) One issue that re-occurs is the description of ‘oogenesis genes.’ The authors show that genes whose expression is enriched during oogenesis are actually expressed during spermatogenesis. This is a surprising and, as they point out in the results section (Pg 5 Ln 137), ‘unexpected.’ The authors need to point out more consistently that this was not expected based on other studies (or explain about the nature of the ‘enrichment’ in oogenesis). In particular:

-Abstract Pg 1 Ln 21: “Here we report that in *Caenorhabditis elegans* sperm, the genome is packaged in nucleosomes and carries a histone-based epigenetic memory of genes expressed during spermatogenesis, which includes genes well known for their expression during oogenesis.”

Change to (suggested wording): “Here we report that in *Caenorhabditis elegans* sperm, the genome is packaged in nucleosomes and carries a histone-based epigenetic memory of genes expressed during spermatogenesis, which unexpectedly includes genes well known for their expression during oogenesis.”

We added “unexpectedly”, as suggested.

-Introduction Pg 3 Ln 59: “Oogenesis-enriched genes are marked in sperm with active marks as a result of their transcription during spermatogenesis.”

Change to (suggested wording): “We also found genes previously found to have enriched expression during oogenesis are also transcribed during spermatogenesis and thus bear active marks as a result of their transcription.”

We modified the sentence to “We found that genes previously shown to have enriched expression during oogenesis are also transcribed during spermatogenesis and thus bear active marks as a result of their transcription.”

2) There is another confusion about the oogenesis gene class. On Pg 5 Ln 11 They define the sex-independent class as those genes that are “expressed in both spermatogenic and oogenic germlines - these genes have “the active modification H3K36me3 and lack the repressive modification H3K27me3.”

However, later, the authors find that many “oogenesis genes” are actually transcribed in spermatogenesis (Pg 6 Ln147). However, these genes bear “H3K36me3, and 75% with H3K4me3, and are devoid of H3K27me3”

This brings up the following questions:

- It is not clear from the data cited (Fig. 3d, Supplementary Fig. 3c, Methods) how this statement is true: “we analyzed our germline RNA-seq data and found that 73% of oogenesis genes (82% of those bearing the active mark H3K36me3 in sperm) are indeed transcribed in spermatogenic germlines.” For example, what subset in Fig 3d is the 73%? Is there some information about the enrichment level of expression that would help clarify this?

We defined genes with RPKM > 15 as ‘expressed’ in Methods. This cut-off is shown in Fig. 3d as gray lines at $y=1.2$ and $x=1.2$ ($\log_{10}(15)=1.2$). 73% of pink dots (oogenesis genes) are located above the $y=1.2$ line (i.e. have RPKM in male germlines >15). We modified that sentence in Results to clarify that point.

- Since “oogenesis genes” are actually transcribed in sperm, are they in the sex-independent class? Are the oogenesis genes and sex-independent genes distinct or overlapping classes? It is unclear how the ‘oogenesis’ class relates to some of the other classes defined because those other classes were defined by this study, while the oogenesis class seems to be defined only by other studies (Pg 20 Ln 539). Did the authors find oogenesis-specific genes that have different markings? This is not meant to ask for further analyses or experiments— it is asking for clarification about the oogenesis gene class and how it is differentiated from the sex-independent gene class.

Oogenesis genes and sex-independent genes were defined differently and for the most part are distinct classes. ‘Oogenesis genes’ were defined by Reinke et al. (2004) as genes that have greater than 2-fold enriched expression ($p < 0.01$) in worms with oogenic germlines vs. worms with spermatogenic germlines (see our Supplementary Fig. 3c). We defined ‘sex-independent genes’ as genes with no significant differential expression between spermatogenic and oogenic germlines (Methods). However, we do see ~30% overlap between oogenesis genes and sex-independent genes. As mentioned in Supplementary Fig. 3 legend, we were not able to identify ‘oogenesis-specific genes’ with the same confidence cut-off as we used to identify ‘spermatogenesis-specific genes’ (Methods). To clarify this point, we added a sentence in the main text: “Indeed, we were not able to identify a high-confidence set of ‘oogenesis-specific genes’ (i.e. expressed only in oogenic germlines but not in spermatogenic germlines) using the same criteria we used to identify ‘spermatogenesis-specific genes (Supplementary Fig. 3c, Methods).”

- Are the markings of oogenesis genes and sex-independent genes distinct? Do the sex-independent class of genes also have H3K4me3?

Histone markings of oogenesis genes and sex-independent genes are very similar in sperm, oocytes, and early embryos (Fig. 2c,d and Fig. 3b,c). Yes, the sex-independent class also has H3K4me3.

3) Pg 3 Ln 72: “We found that sperm retain nucleosomes across all six chromosomes, comparable to early embryos.”

To clarify ‘comparable’ because there is both presence or gross distribution (which I think the authors mean) and levels, which is what is referred to about X in the latter half of the sentence. Are both the levels and distribution comparable? In Supplemental Figure 2b: The ‘density’ is not well explained – a brief explanation would be helpful to more quickly clarify these issues for the reader – density, coverage, distribution, and levels should be clarified.

Fig. 1 and Supplementary Fig. 2 show that the presence and gross distribution are comparable. We changed the main text to clarify: “We found that sperm retain nucleosomes across all six chromosomes, comparable in presence and gross distribution to early embryos ...”

We expanded the legend for Supplementary Fig. 2b to clarify ‘density’.

4) Pg 6 Ln 161: “We conclude that sperm carry a histone-based epigenetic memory of spermatogenesis gene expression, which includes genes well known for their expression during oogenesis.”

This conclusion does not point out that this ‘epigenetic memory’ marking is distinct – that these ‘oogenesis genes’ carry only active marks and not the repressive H3K27me3 mark. This is in contrast to the spermatogenesis-genes that have the multivalent mark. To be clear, it would be helpful in this conclusion point to be specific that the marking is different but each still constitutes a memory of gene expression during spermatogenesis.

We modified the sentence to read: “We conclude that sperm carry a histone-based epigenetic memory of spermatogenesis gene expression, which includes 1) spermatogenesis genes bearing multivalent marking and 2) genes well known for their expression during oogenesis bearing active marking.”

5) Pg 7 Ln 181: Because MES-3 has been shown to be important for silencing X chromosomes in the hermaphrodite germline, can the authors note whether or not the genes that show differential expression in the mes-3 mutants are X genes?

In the main text, we mentioned that “The germlines of M+P- offspring displayed mainly up-regulation of somatic genes and genes on the X chromosome, as previously documented in M+Z- (Maternal load of MES protein, no Zygotic synthesis) mes mutant germlines²³.” X-linked genes were not up-regulated in mes-3 male germlines, because the X chromosome in males has an additional mechanism of repression (namely H3K9me on the unpaired X chromosome, Gaydos et al. 2014). We did not add that point to the text.

6) A general note: The authors use “sperm” throughout the manuscript. The cells isolated by the methods in this manuscript are technically unactivated spermatids that have not developed motility apparatuses to form active spermatozoa. An issue with the use of this term is that sperm used in studies from other organisms are usually spermatozoa. It is highly unlikely that this affects the results and conclusions of this work because in *C. elegans*,

sperm-specific chromatin remodeling occurs concurrent to meiotic divisions and before activation. Nonetheless, the distinction between the cell types should be explained so that other researchers are aware of the differences of *C. elegans* and vertebrate sperm biology. To do this, include in the methods section (Pg 11 Ln 284 after “adapted from.”) something like “These protocols isolate highly enriched populations of spermatids, which have compacted nuclei that have completed meiotic division but have not developed motility structures to form motile spermatozoa. Because these cells have undergone sperm-specific DNA compaction, we refer to these spermatids as mature sperm throughout the text.”

Thank you for the great suggestion, which we incorporated into Methods.

7) Supplemental Figure 1: add scale bars

Scale bars were added.

Reviewer #3 (Remarks to the Author):

All of my concerns about the previous submission have been addressed in this revised submission.